# Dynamic network biomarker indicates pulmonary metastasis at the tipping point of hepatocellular carcinoma

Biwei Yang[1], Meiyi Li[2,3], Wenqing Tang[1], Weixin Liu[3,4], Si Zhang[5], Luonan Chen[3,4] & Jinglin Xia[1,2]

Developing predictive biomarkers that can detect the tipping point before metastasis of hepatocellular carcinoma (HCC), is critical to prevent further irreversible deterioration. To discover such early-warning signals or biomarkers of pulmonary metastasis in HCC, we analyse time-series gene expression data in spontaneous pulmonary metastasis mice HCCLM3-RFP model with our dynamic network biomarker (DNB) method, and identify CALML3 as a core DNB member. All experimental results of gain-of-function and loss-of-function studies show that CALML3 could indicate metastasis initiation and act as a suppressor of metastasis. We also reveal the biological role of CALML3 in metastasis initiation at a network level, including proximal regulation and cascading influences in dysfunctional pathways. Our further experiments and clinical samples show that DNB with CALML3 reduced pulmonary metastasis in liver cancer. Actually, loss of CALML3 predicts shorter overall and relapse-free survival in postoperative HCC patients, thus providing a prognostic biomarker and therapy target in HCC.

[1] Liver Cancer Institute, Zhongshan Hospital, Fudan University, 180 Fenglin Road, Shanghai 200032, China. [2] Minhang Branch, Zhongshan Hospital, Fudan University/Institute of Fudan-Minhang Academic Health System, Minhang Hospital, Fudan University, 170 Xinsong Road, Shanghai 201199, China. [3] Key Laboratory of Systems Biology, CAS Center for Excellence in Molecular Cell Science, CAS Center for Excellence in Animal Evolution and Genetics, Innovation Center for Cell Signaling Network, Shanghai Institute of Biochemistry and Cell Biology, Chinese Academy of Sciences, University of Chinese Academy of Sciences, 320 Yueyang Road, Shanghai 200031, China. [4] School of Life Science and Technology, ShanghaiTech University, 100 Haike Road, Shanghai 201210, China. [5] Key Laboratory of Glycoconjugate Research Ministry of Public Health, Department of Biochemistry and Molecular Biology, Shanghai Medical College, Fudan University, 130 Dong'an Road, Shanghai 200032, China. Biwei Yang, Meiyi Li and Wenqing Tang contributed equally to this work. Correspondence and requests for materials should be addressed to L.C. (email: lnchen@sibs.ac.cn) or to J.X. (email: xiajinglin@fudan.edu.cn)

Hepatocellular carcinoma (HCC) is the third leading cause of cancer-related deaths globally[1]. The high mortality rate results from late presentation at advanced stages, high incidence of tumour metastasis, and tumour recurrence after surgical resection[2]. Generally, HCC is prone to both intrahepatic and extrahepatic metastasis. Extrahepatic metastasis has been reported to occur in 13.5–42% of HCC patients[3,4]. The median survival time and 1-year survival rate of HCC patients with extrahepatic metastasis are only 4.9–7 months and 21.7%–24.9%[3,5], respectively. The most common site of metastasis is lung[6,7]. Metastasis is a nonlinear (i.e., generally irreversible) and dynamic process involving cancer cell motility, intravasation, transit in the blood or lymph, extravasation, and growth at a new site[8]. Understanding the molecular mechanisms of this irreversible HCC metastasis at a network level is of great importance, both for preventing the initiation of metastasis in early HCC patients and for developing therapeutic strategies in advanced HCC patients.

One invariable feature of the metastatic process is deregulated gene expressions and dysfunctional interactions, which dynamically affects sequential stages of tumour cell invasion, organ tropism, and growth at distant sites[9]. Various oncogenes and tumour suppressors forming networks or pathways are involved in the metastatic process. Pathway-based approaches and functional experimental studies have been adopted in identifying the dysfunction of different signalling cascades in HCC metastasis (e.g., insulin-like growth factor (IGF), mitogen-activated protein kinase (MAPK), phosphatidylinositol-3 kinase (PI3K)/AKT/ mammalian target of rapamycin (mTOR), and WNT/β-catenin)[10] and disease-related biomarkers. Although some of these biomarkers are effective in identifying HCC patients who are in a metastasis state, it is difficult to pinpoint the critical state or tipping point before metastasis initiation (i.e., to identify HCC patients who are in a metastasis-imminent state) for early diagnosis. Specifically, HCC progression can be divided into three stages: non-metastatic state, pre-metastatic state (i.e., a critical state/tipping point, and still a reversible state), and metastatic state (a generally irreversible state). Clearly, there is a phase transition just after the pre-metastasis state that leads to a drastic (irreversible) change in phenotype[11,12]. Generally, there are significant differences between non-metastatic and metastatic states in terms of gene expression, which is why we can find molecular biomarkers to distinguish the two states. However, statically there is no clear difference between non-metastatic and pre-metastatic states, because the pre-metastasis state is really a part of the non-metastatic state. Thus, "traditional" molecular biomarkers fail to distinguish them or fail to identify HCC patients in the pre-metastasis state.

Recently, new high-throughput omics technologies (e.g., microarrays and deep sequencing), sophisticated animal models (e.g., mosaic cancer mouse models with the use of transposons for mutagenesis screens), loss-of-function (e.g., CRISPR/ Cas9 system) and gain-of-function (e.g., Tet-on inducible system) studies have opened the field to new strategies in oncogene and tumour suppressor discovery, in particular, for studying the pre-metastatic state and the critical transition problem from the perspectives of both network and dynamics[11–17]. Actually, in contrast to no statically significant difference, it has been shown that dynamically there is significant difference between non-metastatic (or normal) and pre-metastatic (or critical) states, which can be explored to develop dynamic biomarkers (rather than the traditional static biomarkers) for predicting the pre-metastatic (or critical) state.

In this work, we adopted our mathematical method, i.e., the dynamic network biomarker (DNB) model, to identify the pre-metastatic state or tipping point by exploring dynamic and network information of omics data from both animal models and

clinical samples[11,12,15,17]. Actually, the DNB model has been also recently applied to analyse other complex biological processes by many other researchers, e.g., successfully identifying the tipping points of cell fate decision[13,14] and studying immune checkpoint blockade[16]. Specifically, we obtained DNB genes that not only signalled the pre-metastatic state but also were strongly related to key molecules of HCC metastasis, by analysing the three DNB statistical conditions of the critical state derived from nonlinear dynamic theory[11,13–15,17]. Compared with traditional biomarkers detecting the metastatic state based on differential expression of molecules, a major advantage of the DNB method is to identify the pre-metastatic state or critical state (tipping point) just before the irreversible transition to metastasis state in tumour progression. Furthermore, we discovered several DNB members as predictive biomarkers that were shown to play a precursor role in initiating metastasis[11,12,15,17] in both animal models and patient samples.

Specifically, we use xenografts and a spontaneous pulmonary metastasis mouse model of HCCLM3 to detect the tipping point of HCC metastasis initiation and its predictive biomarkers[18]. The model with a transplanted HCC cell line, i.e., HCCLM3-RFP[18] of high metastatic potential, is labelled with a stable fluorescent protein, and is effective in resolving and quantifying the dynamics of tumourigenesis and metastasis. By analyses of time-series whole-genome expression patterns of orthotopic liver tumour samples, based on the DNB model, we find remarkable changes at the third week after orthotopic transplant (the critical or pre-metastatic state), which was consistent with the results of circulating tumour cells (CTC) detection.

In this study, we demonstrate a new way to predict the critical transition in metastasis by DNB method, and provide new insights into the molecular pathology of HCC pulmonary metastasis from the perspectives of dynamics and network. In particular, we find that CALML3, as a DNB member or dynamic biomarker, could not only quantify the early-warning signals of metastasis initiation but also effectively predict outcomes in HCC patients.

## Results

**Tumourigenesis and pulmonary metastasis in xenograft model.** The tumour microenvironment plays an essential role in tumour growth and metastasis in tumour progression[19]. To simulate tumour growth and metastasis in livers of HCC patients, we used the spontaneous pulmonary metastasis mouse model, HCCLM3-RFP, which involved the orthotopic transplanted human HCCLM3 cell line labelled with a stable fluorescent protein[18]. Due to the high metastatic potential of HCCLM3, this cell line has been used frequently in studies seeking to identify biomarkers and to examine mechanisms in the pulmonary metastasis of human HCC[20–22]. From experience in previous work[18], we created a xenograft model according to the flowchart shown in Fig. 1a. Continuous observation of the HCCLM3-RFP model under fluorescence stereomicroscope revealed that tumourigenesis occurred in three nude mice at the second week after subcutaneous injection. To create the orthotopic xenograft model, we implanted pieces of these tumours into the liver of several dozen nude mice when the diameter of the subcutaneous tumours was larger than 1 cm in the three subcutaneous xenograft mice.

We observed that, hepatic tumours in orthotopic xenograft HCCLM3-RFP mice grew gradually from the second week to the fifth week after orthotopic implantation in the primary liver tissue, whereas spontaneous pulmonary metastasis occurred only at the last time point (the fifth week after orthotopic implantation) (Fig. 1b). The fluorescence observation of the liver tumours and lung metastatic foci at the fifth week after orthotopic

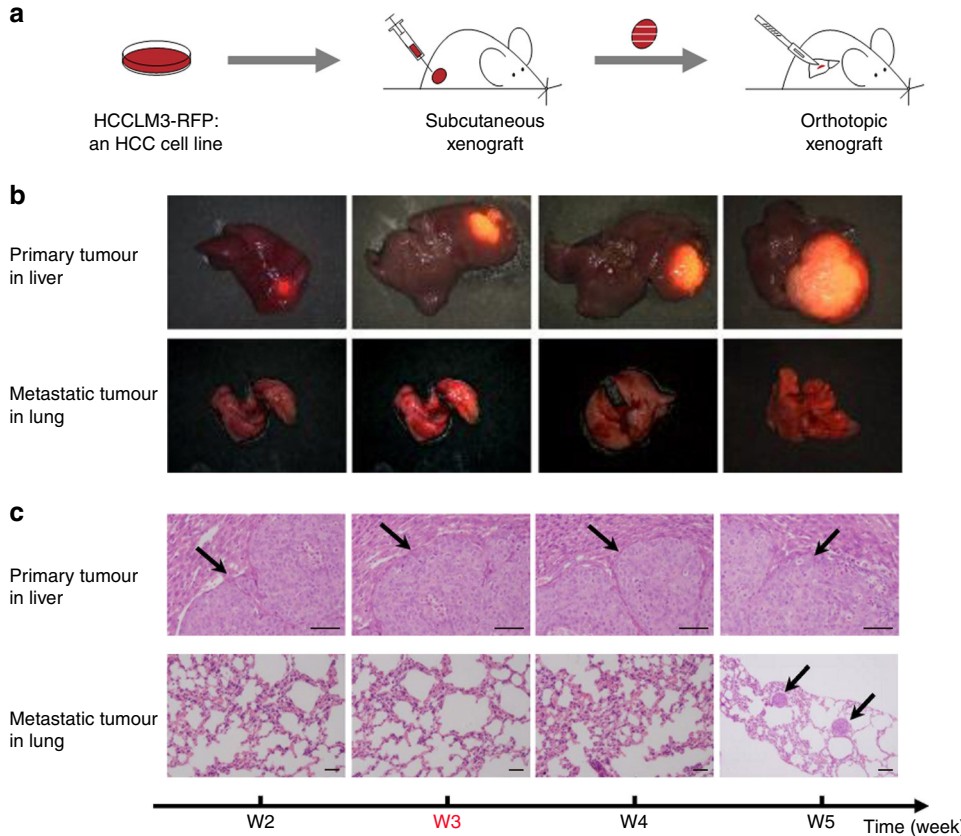

**Fig. 1** Xenograft mouse model of HCCLM3-RFP for pulmonary metastasis of hepatocellular carcinoma (HCC). **a** The flowchart showed the process of creating the xenograft mouse model, HCCLM3-RFP. Stage I, HCCLM3-RFP cell lines were cultured and amplified. Stage II, each of three nude mice were injected subcutaneously with $1 \times 10^7/0.2$ ml HCCLM3-RFP cells in the right upper flank region to establish the subcutaneous xenograft model. Stage III, after 30 days the subcutaneous tumour was removed and cut into pieces and implanted orthotopically into the livers of other mice. **b** The series of images showed the growth and metastasis of transplanted hepatic tumours in each of the orthotopic xenograft mice from the second week after orthotopic implantation to the fifth week. The transplanted tumours in the mice livers increased gradually from the second week, and spontaneous pulmonary metastasis of HCCLM3-RFP could be observed only at the fifth week. **c** Histological images of tissue sections stained with haematoxylin and eosin (H&E) at each time point validated the progression of hepatocarcinogenesis and pulmonary metastasis (scale bar, 100 μm)

implantation were confirmed by haematoxylin and eosin (H&E)-stained histopathology (Fig. 1c). Thus, we chose the second, third, fourth, and fifth weeks after orthotopic implantation as observation points (four time points) to collect liver tumours of five orthotopic xenograft mice at each time point and to assess the whole-genome expression.

**Dynamic gene expressions during HCC pulmonary metastasis.** Generally, a whole-genome expression profile is expected to reflect the instant state of tumour. Thus, to characterise state-specificity during the progression of HCC pulmonary metastasis, we identified 13,247 differentially expressed genes (DEGs) by multiple comparisons with false discovery rate (FDR) adjustment ($P < 0.05$; Supplementary Data 1). Then, after performing unsupervised hierarchical clustering based on the identified DEGs (Fig. 2a), we found that all primary tumour samples at the four time points (W2, W3, W4, W5) could be clustered roughly into two independent groups (i.e., one primarily including all samples at the second week after orthotopic implantation, two samples at the third week and one sample at the fifth week; the other including all samples at the fourth and fifth weeks (except W5-a) and three samples at the third week). This suggested that these hepatic tumours, across the four time points, reflected the progression from a non-metastatic state to a metastatic state. Indeed,

the five tumour samples from the third week after orthotopic implantation were not clustered into one group (red) but, instead, scattered in both groups. This implied that tumour cells at this time point might be in the critical period of a pulmonary metastatic or a pre-metastatic state.

To further understand the dynamic changes at the gene expression level during the metastasis progression, we classified all DEGs into six patterns (Cluster 1,…,Cluster 6) using Mfuzz[23] (Fig. 2b). Genes in Clusters 1 and 2 were continuously and monotonically downregulated and upregulated from the second to fourth weeks after orthotropic implantation, respectively. Genes in Clusters 3 and 4 were significantly downregulated and upregulated only during the second and third weeks, respectively, whereas genes in Clusters 5 and 6 were significantly downregulated and upregulated only during the third and fourth weeks, respectively. From these results, we concluded that disease progression was not gradual and monotonic, but passed through nonlinear and drastic transitions at certain points[11]. In fact, we found that there were significant changes in gene expression between the third and fourth weeks, but no significant difference between the fourth and fifth weeks after implantation; this indicated that the tipping point or critical state may occur in the third week and that hepatic tumour cells may metastasize in the fourth week.

**Identifying tipping point of HCC pulmonary metastasis by DNB**. Based on nonlinear dynamic theory, the appearance of a DNB implies the tipping point or the emergence of the critical transition just before metastasis initiation[11–17] (Fig. 2c). Generally, DNB is a group of molecules with strong correlations and fluctuations, which differ from the molecules with differential expression used widely in "traditional" methods[11,12,15,17]. We can identify the DNB group or dominant group among all observed molecules according to the three criteria or conditions (DNB analysis in Methods, Fig. 2d and Supplementary Fig. 1 for details)

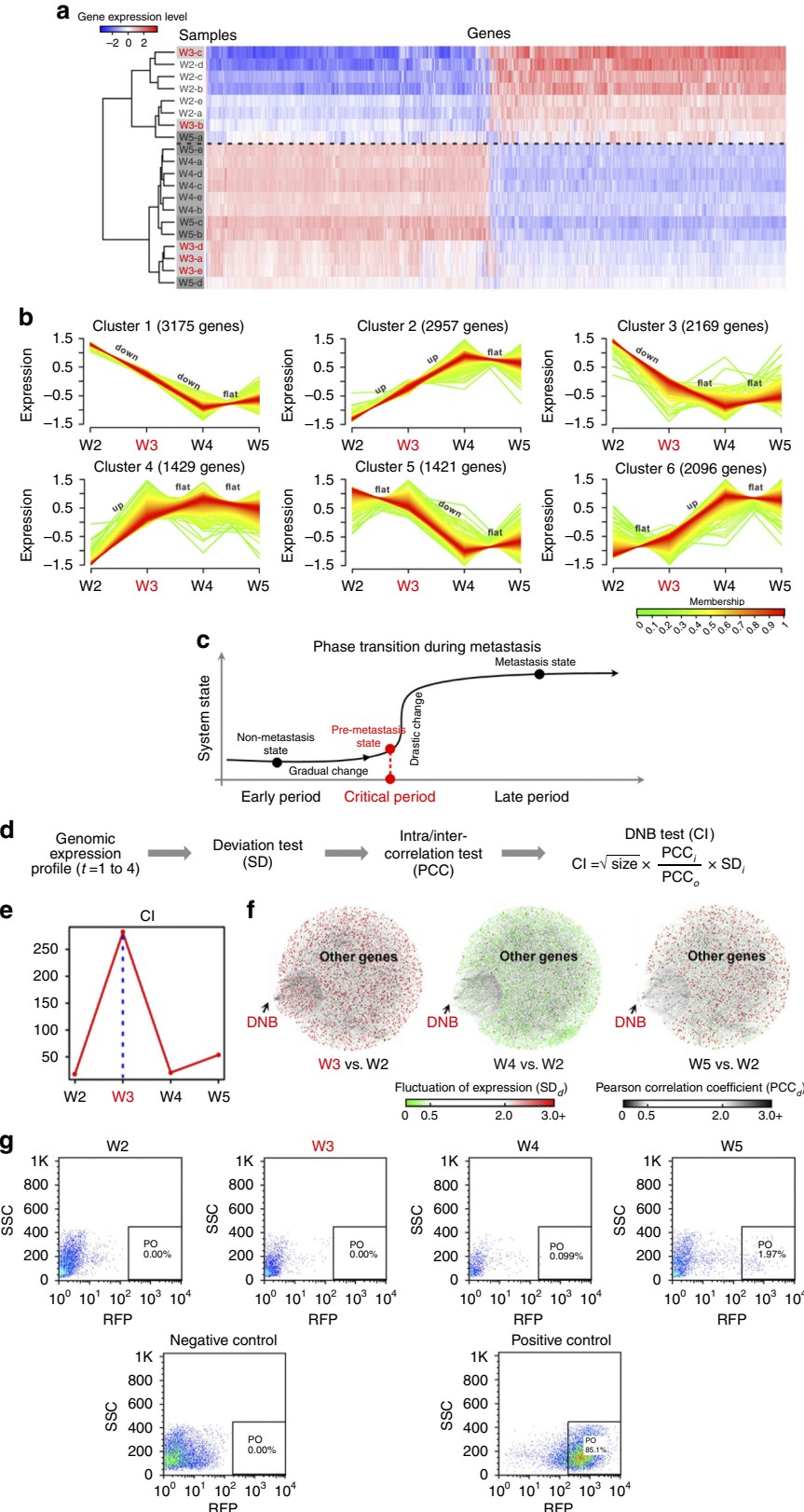

when the system state gradually approaches a tipping point. Intuitively, DNB theory indicates that the appearance of a group of molecules with strongly collective fluctuations within the observed data implies the tipping point or an imminent transition in the disease, and thus those molecules or DNB members are the predictive biomarkers. Clearly, DNB is dynamic biomarkers to predict diseases, in contrast to traditional static biomarkers (e.g., molecular biomarkers) to diagnose diseases.

After analysing whole-genome expression profiles from the tumour samples of orthotopic xenograft mice at four time points, we found a strong signal of the critical state before metastasis initiation by CI (DNB analysis in Methods) at the third week after implantation (Fig. 2e and Supplementary Fig. 2). This result was consistent with the morphological alterations in orthotopic xenograft mice (Fig. 1b, c) and the dynamics of gene expressions (Fig. 2b). Meanwhile, we obtained the corresponding DNB members, which were composed of 334 genes or proteins (Supplementary Data 2). The analysis of biological functions in networks also indicates the systematic interplay of multiple molecules[24–26]. To show visually the dynamics of DNB as a network during the progression of pulmonary metastasis for primary tumour cells, we integrated the human molecular interactions from KEGG[27], BioGRID[28], and TRED[29], and then weighted nodes and links individually from this entire network according to the SDs (standard deviations) of gene expression and the PCCs (Pearson correlation coefficients) of pair-wise genes vs. those at the original time point (Fig. 2f). Compared with the whole molecular network, we found that this group of DNB genes could specifically signal the tipping point or the pre-metastatic state, at the third week (Fig. 2f). Note that DNB genes were identified only based on the three DNB criteria. Thus, there was no requirement on differential expression for those DNB genes. In other words, a DNB gene may be a DEG gene or not.

**Validating the tipping point of metastasis by CTCs**. Although we have observed pulmonary metastasis by fluorescence stereo-microscopy at the fifth week after orthotopic implantation, the previous results based on gene expression levels indicated that hepatic tumour cells may metastasize from the fourth week. Breaking away from the primary solid tumour into the circulating blood is an important step for distant metastasis to the tumour cells[30]. To more exactly identify the time point of metastasis initiation experimentally for primary tumour cells in the liver, we used flow cytometry to measure CTCs[31], i.e., HCCLM3-RFP cells, in plasma from the peripheral blood of orthotopic xenograft mice at each time point. The result showed that CTCs only appeared in peripheral blood of the orthotopic xenograft mice at the fourth and fifth weeks after orthotopic implantation (Fig. 2g), which was

consistent with the previous conclusion from gene expression and DNB analysis, that primary tumour cells metasazise at the fourth week after implantation. The third week was likely to be the pre-metastatic stage or the tipping point just before metastasis.

**CALML3 plays a key role based on DNB ranking**. To further understand the underlying roles of DNB genes in metastasis initiation for primary tumour cells, we ranked comprehensively the DNB genes according to the criteria with four priorities, i.e., importance in the networks, importance in functional pathways, differential patterns, and dynamic patterns (see Ranking scheme for DNB members in Methods), and selected CALML3 as the top one for further functional study (Fig. 3a, Supplementary Data 2). Certainly, based on different considerations, there might be other ways to rank the genes.

**Loss of CALML3 is associated with high metastatic potential**. To explore relationship between dysfunctional CALML3 and metastasis potential in liver cancer cells, we studied the expression profile of CALML3 in human hepatocyte cell line THLE-3 and seven hepatoma cell lines with different metastatic potentials, including cells with no or low metastatic potential (HepG2, Bel-7402, SMMC-7721 and PLC/PRF/5) and serial cell lines with medium or high metastatic potential (MHCC97L, MHCC97H and HCCLM3). MHCC97L, MHCC97H and HCCLM3 (metastatic potential: MHCC97L < MHCC97H < HCCLM3) were established from the same parent HCC cell line, MHCC97, at Liver Cancer Institute, Zhongshan Hospital, Fudan University[32]. Compared with the high expression of CALML3 in THLE-3, CALML3 was significantly downregulated in most hepatoma cell lines, especially in cells (MHCC97L, MHCC97H and HCCLM3) with relatively high metastatic potential, and moderately expressed in hepatoma cells (HepG2, SMMC-7721 and PLC/PRF/5) with low metastatic potential (Fig. 3b). Notably, there was step-wise decreased CALML3 expression in MHCC97L (relatively high), MHCC97H (medium) and HCCLM3 (relatively low), which was negatively correlated with their metastatic potential. The expression level of CALML3 in hepatoma cell lines with high metastatic potential was remarkably lower than in those with lower metastatic potential, confirming that CALML3 was a metastasis suppressor.

**CALML3 is a HCC metastatic suppressor in vitro and in vivo**. To further explore the role of CALML3 in HCC metastasis, we performed gain-of-function and loss-of-function studies to detect the effect of CALML3 on oncogenic behaviours of hepatoma cells including cell growth, migration and invasion. Considering that MHCC97L and HCCLM3 (metastatic potential: MHCC97L <

**Fig. 2** Dynamic changes in the primary tumour in terms of gene expression during the progression of pulmonary metastasis. **a** A heatmap shows a clustering analysis by unsupervised hierarchical clustering with Euclidean distance based on differentially expressed genes (DEGs). The primary tumours in mice at the third week after orthotopic transplant in terms of gene expression were not grouped together but scattered, indicating that the state at the third week (W3) is special and different from the other time points. **b** The series of diagrams illustrates the patterns of dynamic changes in DEGs during the progression of pulmonary metastasis, using Mfuzz. **c** A schematic diagram illustrates a phase transition during metastasis. HCC progression can be divided into three stages: non-metastatic, pre-metastatic (i.e., the critical state), and metastatic. Generally, there are significant differences between the non-metastatic and metastatic states; however, there is no clear difference between the non-metastatic and pre-metastatic states, because the pre-metastatic state is in fact part of the non-metastatic state, so the traditional biomarkers fail to distinguish them due to their static properties. The critical transition after the pre-metastatic states changes the state of the biological system qualitatively, and thus plays a key role in biological processes. **d** The flowchart shows the computational algorithm of our mathematical model to identify dynamic network biomarkers (DNBs), signalling the pre-metastatic state. **e** The graph illustrates that the critical transition for the primary tumour occurs at the third week after the orthotopic transplant, according to CIs over all time points in gene expression profiling. **f** The series of networks show graphically that the three criteria of DNB were satisfied at the third week from dynamic changes in the variations in gene expression and the network structure during metastasis, compared with the entire molecular network. **g** Flow cytometric analysis showed that circulating tumour cells (CTCs), separated from peripheral blood mononuclear cells of the xenograft mice, were observed at the third week after orthotopic transplantation. Wild-type HCCLM3 cells, without red fluorescence, is the negative control and HCCLM3-RFP is a positive control

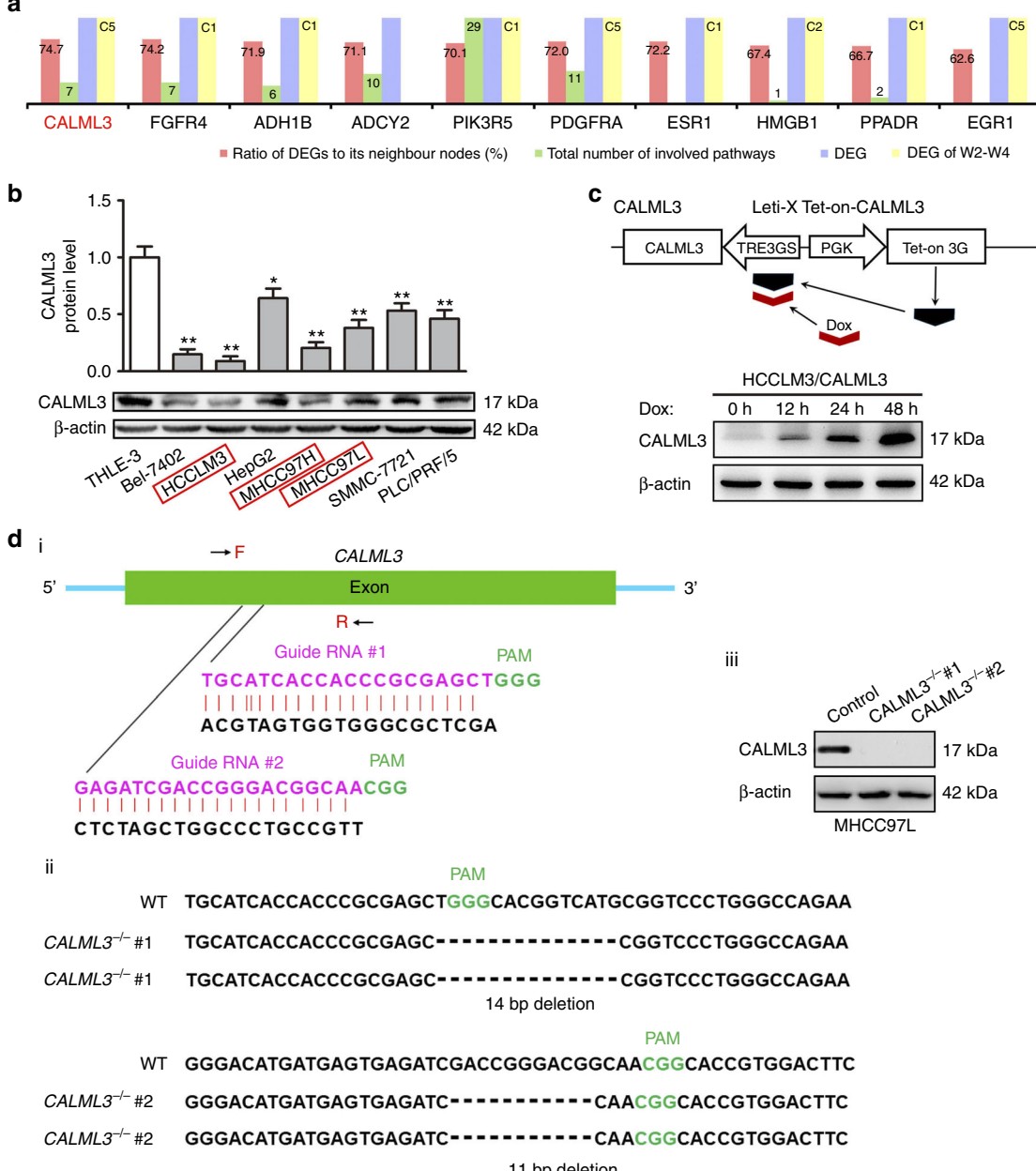

**Fig. 3** CALML3 ranked as one of core DNB members and generation of CALML3 overexpression and knockout cells by Tet-on and CRISPR/Cas9 systems. **a** The diagram shows the DNB ranking, based on the various criteria with the four priorities, during the progression of metastasis. CALML3 was chosen as the top one for further functional tests. Note that the patterns of the change for a DNB gene at gene expression level were labelled as 1 or 0 (on the third and fourth bars). For DEG of W2–W4, C indicates a Cluster (e.g., C5 means that the gene belongs to Cluster 5). **b** The protein expression levels of CALML3 in a normal human liver epithelial cell (THLE-3) and multiple cell lines with different metastatic potentials. The rectangles with red frames showed MHCC97L, MHCC97H and HCCLM3 (metastatic potential: MHCC97L < MHCC97H < HCCLM3), which were established from the same parent HCC cell line, MHCC97. **c** Establishment and characterisation of a doxycycline-inducible CALML3 overexpression system. Upper panel: Schematic of Lenti-X Tet-One inducible expression system to express inducible CALML3. Dox, doxycycline; TRE, tetracycline responsive element. Lower panel: Western analysis of CALML3 in extracts derived from inducible CALML3 cells, either untreated or induced with doxycycline (50 ng/ml) at indicated time points. **d** Generation of CALML3-knockout MHCC97L cell lines (clones of CALML3$^{-/-}$ #1 and CALML3$^{-/-}$ #2 cell lines) using the CRISPR/Cas9 system. (i) Schematic representation of the CALML3-targeting gRNA sequences. Arrows indicate primer positions. PAM protospacer adjacent motif. (ii) Two CALML3$^{-/-}$ cell lines were established from MHCC97L cells. The deleted sequences in the CALML3$^{-/-}$ #1 and CALML3$^{-/-}$ #2 cell lines are presented. Individual colonies were selected and genotyped by genomic DNA sequencing. (iii) Western analysis of CALML3 in extracts derived from control and two CALML3$^{-/-}$ cell lines. β-actin was used as the loading control. Student's $t$-test. *$P < 0.05$, **$P < 0.01$. Error bars in panels are defined by SD (standard deviation)

HCCLM3) were established from the same parent HCC cell line and showed relatively high and low CALML3 expression respectively, we established a doxycycline-inducible CALML3 overexpression system in HCCLM3 and a CRISPR/Cas9-mediated knockout in MHCC97L cells. Treatment with 50 ng/ml doxycycline induced a time-dependent expression of CALML3 protein in HCCLM3/CALML3 cells, detectably as early as 12 h (Fig. 3c). Two CALML3-targeting gRNAs sequences were designed to avoid non-specific effects of the CRISPR/Cas9 system. Confirmation of the genotypes of two CALML3$^{-/-}$ cell lines (CALML3$^{-/-}$ #1 and CALML3$^{-/-}$ #2) was testified by genomic DNA sequencing and western blot (Supplementary Fig. 3 and Fig. 3d).

Cancer is characterised by sustaining proliferative signalling, evading growth suppressors, resisting cell death, enabling replicative immortality, inducing angiogenesis, and activating invasion and metastasis[33]. The result of cell proliferation assay showed that inducible CALML3 expression significantly inhibited cell growth in HCCLM3 cells, while knockout of endogenous CALML3 promoted cell proliferation in MHCC97L cells (Fig. 4a). The results of cell migration and invasion assay showed that compared to the controls, CALML3 overexpression cells displayed markedly decreased ability of migration and invasion, however, CALML3 knockout had an opposite effect on metastasis of HCC cells (Fig. 4b–e). In accordance with assay in vitro, the result of xenografts models established by inducible CALML3 overexpression system in HCCLM3 cells verified that overexpression of CALML3 significantly diminished tumourigenic capacity and pulmonary metastasis. Inducible CALML3 expression group exhibited less weight and volume of tumours, as well as less pulmonary metastasis nodules and incidence than the control (Fig. 4f–i). In addition, to further verify the role of CALML3 as a tumour suppressor, we overexpressed CALML3 in HCCLM3 cells with high metastatic potential and decreased CALML3 expression in HepG2 cells with low metastatic potential by lentiviral-mediated expression and RNA interference, respectively (Supplementary Fig. 4a). The results also showed that overexpression of CALML3 remarkably inhibited cell proliferation, cell migration and cell invasion (Supplementary Fig. 4b–f), whereas downregulation of CALML3 had the opposite effect. We also created xenografts models of the control HCCLM3 and CALML3 overexpression HCCLM3 cells with fluorescent proteins as above (Fig. 1a); CALML3 overexpression significantly inhibited tumour growth and tumour pulmonary metastasis (Supplementary Fig. 4g, h). Altogether, the results of these functional assays in vitro and in vivo suggested that CALML3 was a suppressor gene in HCC tumourigenesis and metastasis.

**Rewiring of CALML3-subnet before and after the tipping point.** Metastasis has been considered a complex and dynamic process, involving multiple genes working in a concerted manner[34,35]. Based on previous work, DNB members have been considered to be leading factors, situated at important positions (e.g., upstream) of pathways that regulate key disease-associated processes during the initiation and development of complex diseases[11,15,17]. Thus, we analysed CALML3-associated regulations and functions (or network rewiring) to systematically explore the role of CALML3 in the metastatic process at a network level. According to the knowledge-based molecular interactions of human biology, we integrated CALML3 and its 72 neighbouring genes into a CALML3-centred network and weighted the nodes of this network according to the z-score transformed data of their real gene expression across the four time points (Fig. 5a). The expression levels of 54 neighbouring genes (three-quarter of all CALML3 neighbouring genes,

excluding eight previously undetected genes in the whole-genome expression profile) all reversed, from high (low) to low (high) levels before and after the tipping point of metastasis initiation (the third week after orthotopic implantation). This suggested that the tumour suppressor CALML3 played a critical role in metastasis initiation, directly or proximally regulating these reversing DEGs at a molecular network level.

**CALML3 is situated upstream of metastasis-related pathways.** Next, in examining the relationship between the initiation of metastasis and biological functions enriched by CALML3 and its 53 inversing DEGs before and after the critical period of metastasis initiation, we found that most typical cancer-related pathways (e.g., apoptosis, HIF-1) were enriched (Fig. 5b, Supplementary Data 4). In particular, the enriched functions in which CALML3 participated directly were related primarily to signal transduction (e.g., phosphatidylinositol, cAMP, Ca$^{2+}$ and cGMP-PKG). These enriched pathways belong to typical cancer-associated signals resulting in cell growth and division[36,37] as well as cell adhesion and migration[37,38], and play key roles in metastasis initiation. Here, inversing DEGs refer to those genes whose expressions changed significantly from low (high) to high (low), before and after the tipping point of metastasis (i.e., the third week). The dynamics of the functional phenotypes of CALML3 and its inversed DEGs indicated the complexity and time-dependence of dysfunctions in metastasis-related biological processes during metastasis initiation (Fig. 5b). Most of the pathways involved in cell growth and differentiation (e.g., cAMP, cell cycle, and MAPK signalling) and typical cancer-associated mechanisms (e.g., lipid and amino-acid metabolism) were dysregulated before the pre-metastatic state, whereas other pathways involved in the immune response were dysregulated after the pre-metastatic state. Dysfunction in cell adhesion and migration, especially those involving CALML3, occurred throughout the metastasis period.

The above analyses showed that the dynamics resulted proximally from the regulation or dysfunction in CALML3 during the initiation of metastasis. Furthermore, we sought to determine whether CALML3 could regulate well-known metastasis-related genes in a cascading manner. For this, we used a polymerase chain reaction (PCR) array including 84 typical cancer-associated genes (e.g., ones involved in cell adhesion, extracellular matrix (ECM) components, cell cycle, cell proliferation and apoptosis) and five housekeeping genes (as controls), to validate CALML3-regulated DEGs in wild-type HCCLM3 cells (HCCLM3/NC) and CALML3 overexpression HCCLM3 cells (HCCLM3/CALML3) (Supplementary Data 5). The expression levels of 37 genes were regulated by CALML3, of which 32 genes were also seen in our whole-genome expression data (Fig. 5c). Compared with the previous analysis of transcriptomic data, 17 genes, listed on the left of the broken line, were also DEGs; and 20 genes, with green stars, had the same correlation or regulation with CALML3 (the red genes had positive correlations with CALML3, whereas the black ones had negative correlations; Fig. 5c). These results indicated that CALML3 was situated in the upstream of the metastasis-associated pathways, playing an important role in the initiation of metastasis.

**Loss of CALML3 predicts poor prognosis in HCC patients.** We studied the CALML3 expression of a cohort of 270 HCC patients consisting of 100 patients with pulmonary metastasis and 170 without pulmonary metastasis by immunohistochemistry (IHC) (Fig. 6a). Clinical characteristics of patients were shown in Supplementary Table 1. There were significant differences between patients with or without pulmonary metastasis, i.e., CALML3

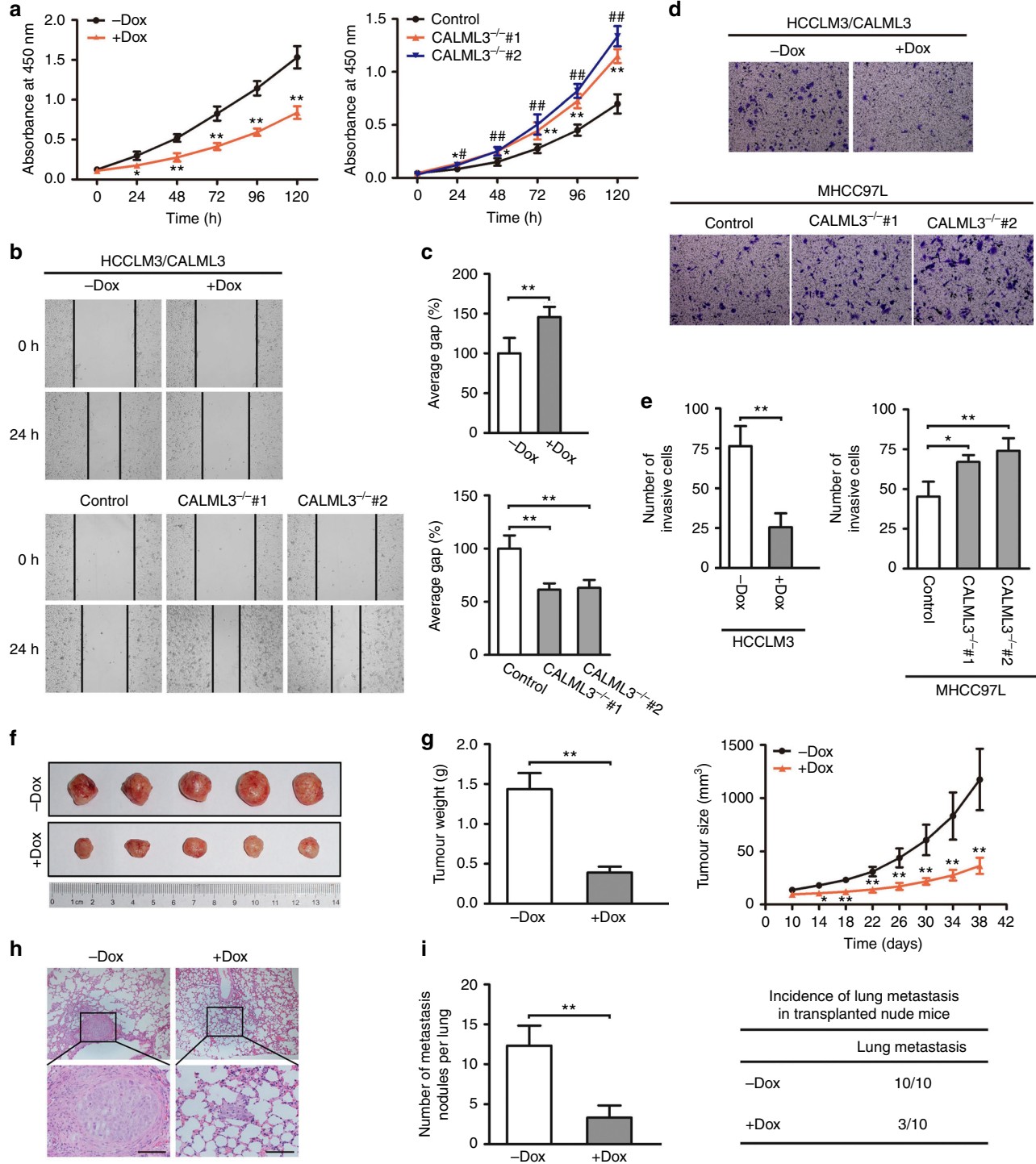

**Fig. 4** CALML3 acts as a HCC metastatic suppressor in vitro and in vivo. **a** CALML3 overexpression in HCCLM3 cells induced by doxycycline (50 ng/ml) for indicated time inhibited cell proliferation, while CALML3 knockout by CRISPR-Cas9 technology promoted cell proliferation. Two clones (CALML3$^{-/-}$ #1 and CALML3$^{-/-}$ #2) targeting CALML3 were designed for functional assays. **b** CALML3 overexpression suppressed cell migration (upper panel) while CALML3 knockout brought the opposite effect (lower panel). **c** Quantification of migration cells in the indicated groups in the wound healing assay. CALML3 overexpression showed obvious suppression of migration abilities in HCCLM3 cells (upper panel) while CALML3 knockout induced significant promotion of migration abilities in CALML3$^{-/-}$ #1 and CALML3$^{-/-}$ #2 MHCC97L cells (lower panel). **d** CALML3 overexpression suppressed cell invasion (upper panel) while CALML3 knockout brought the opposite effect (lower panel). **e** Quantification of invasive cells in the indicated groups in the transwell invasion assay. **f** Tumours from mice implanted with HCCLM3 cells (the control and CALML3 overexpression) in tumour-bearing mouse model. **g** Comparison of tumour weight and size in tumour-bearing mouse model assay. CALML3 overexpression induced by doxycycline showed less tumour weight and significantly slowed tumour growth. **h** Lung tissues from mice implanted with HCCLM3 cells (control and CALML3 overexpression) orthotopic transplantation model were stained with hematoxylin-eosin (scale bar, 100 μm). **i** Comparison of pulmonary metastasis in xenografts models. CALML3 overexpression's induced by doxycycline showed less lung metastatic nodules and reduced incidence of lung metastasis. Dox: doxycycline. Student's *t*-test. *$P < 0.05$, **$P < 0.01$. Error bars in panels are defined by SD (standard deviation)

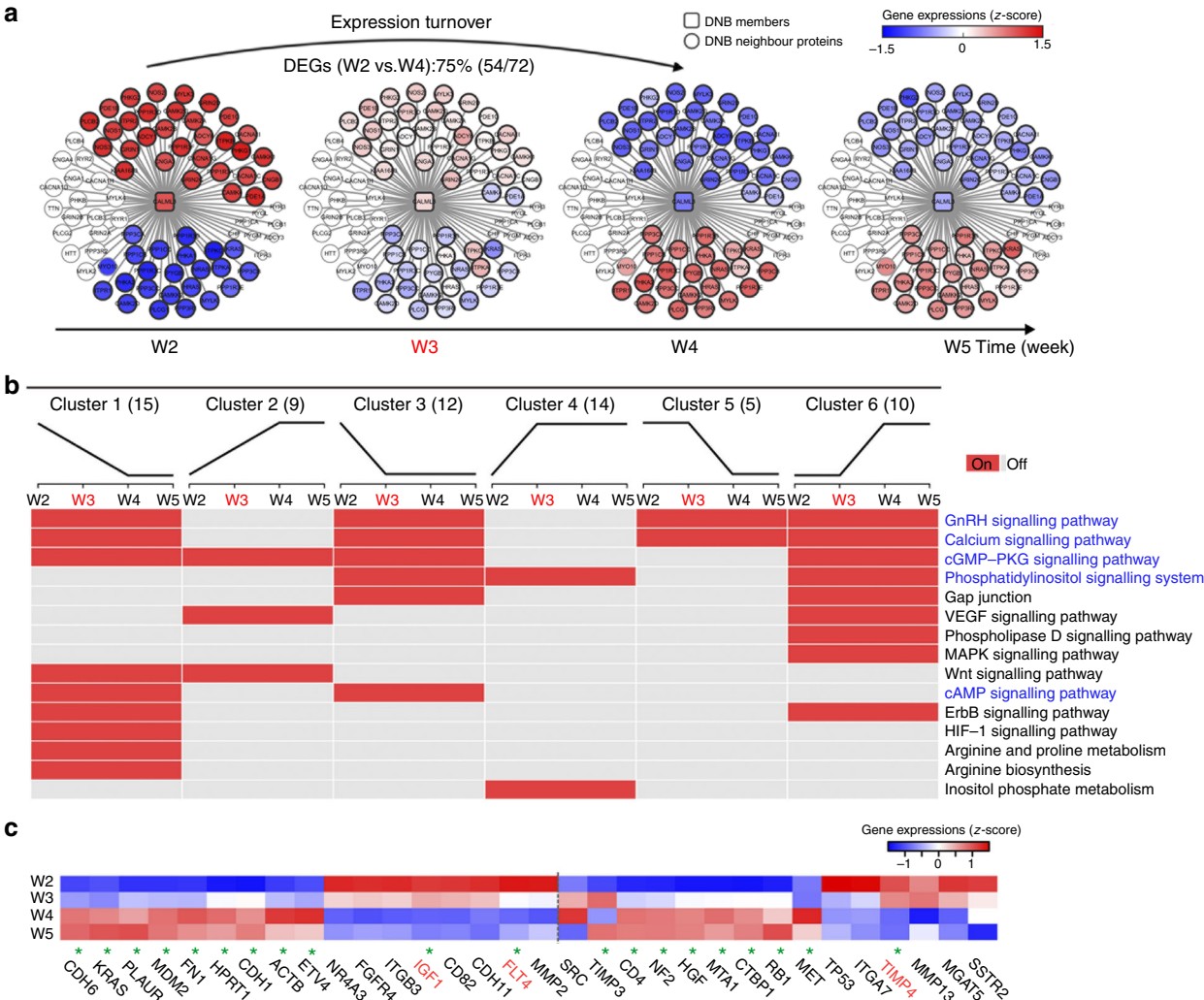

**Fig. 5** Dynamics of CALML3-affected genes in terms of expression and function during the progression of pulmonary metastasis. **a** These series of networks demonstrated the dynamic changes in CALML3 and its neighbouring genes in terms of expression during metastasis. Expression of CALML3 and its 53 neighbouring DEGs changed significantly (or inversed) from low (or high) at week 2 to high (or low) at weeks 4 and 5 (i.e., the expression of those genes inversed before and after the critical period, at week 3). Here, expression turnover means that the expression of the genes changed significantly from low (high) to high (low) before and after the tipping point of metastasis (week 3). **b** A heatmap showed dynamic activity of the related KEGG pathways that involved CALML3 and its neighbouring DEGs in different dynamic patterns. **c** A heatmap illustrated the coexpression of CALML3 and typical cancer-related genes. The 17 genes listed on the left of the broken line were also DEGs in the transcriptomic data; 20 genes labelled by a green star had the same relationships with CALML3 by analysis of both the whole-genome expression profiling and the RT$^2$ Profiler PCR arrays. Genes in red showed positive coexpression with CALML3, whereas genes in black showed negative coexpression with CALML3

expression in tumour tissue, tumour encapsulation, vascular invasion, Barcelona Clinic Liver Cancer (BCLC) stage and pro-thrombin time (Supplementary Table 1). In total, sixteen clinical characteristics (e.g., tumour encapsulation, vascular invasion, tumour number, tumour size, BCLC stage, intratumoural CALML3 expression and so on) were analysed to identify factors associated with overall survival (OS) and relapse-free survival (RFS). The result of univariate analysis showed that intratumoural CALML3 expression, tumour encapsulation, vascular invasion, tumour number, tumour size, Edmondson grade, BCLC stage, ALT (alanine aminotransferase), TB (total bilirubin) and AFP (alpha fetoprotein) were significantly associated with OS in univariate analyses. Intratumoural CALML3 expression, tumour encapsulation, vascular invasion, tumour size, Edmondson grade, BCLC stage and prothrombin time were significantly associated with the RFS (Supplementary Table 2). With a multivariate Cox

regression model, intratumoural CALML3 expression, tumour encapsulation, vascular invasion, BCLC stage and TB were factors related to OS. Intratumoural CALML3 expression, vascular invasion and prothrombin time were factors related to RFS (Supplementary Table 3). Clinicopathologic analysis showed that CALML3 expression was significantly correlated with tumour encapsulation, vascular invasion, tumour size and BCLC stage (Supplementary Table 4). CALML3 expression was remarkably reduced in the tumour tissues (average score, 3.84 ± 3.14), comparing with the corresponding peritumoural liver tissues (average score, 7 ± 3.22) (P < 0.001, Fig. 6a, b). Moreover, CALML3 expression was significantly lower in the tumour tissues of pulmonary metastatic group (average score, 1.94 ± 2.19; 86%, 86/100) than that of non-pulmonary metastatic group (average score, 4.96 ± 3.08; 41.1%, 70/170) (Fig. 6c, d), indicating that loss of CALML3 was a risk factor of pulmonary metastasis of HCC.

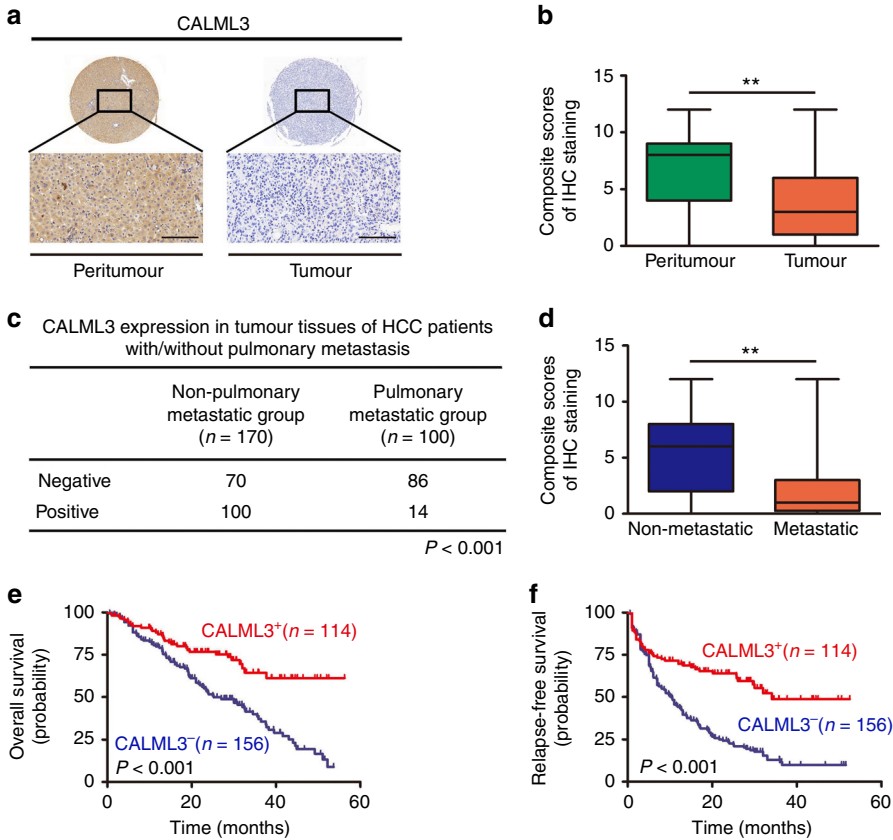

**Fig. 6** CALML3 analysis by immunostaining in tumour tissue slices from patients. **a** Histological images of tissue specimens from HCC patients show different patterns of CALML3 staining (scale bar, 50 µm). CALML3 staining was primarily in the cytoplasm. **b** Scores of immunochemistry staining of CALML3 in 270 pairs of tumour tissues and peritumour tissues. The horizontal lines in the box plot (minimum to maximum) represent the median and the interquartile ranges. Error bars represent mean ± SD. IHC scores were compared by Student's t-test, **P < 0.01. **c** The intratumoural CALML3 expression was significantly lower in the pulmonary metastatic group than that in the non-metastatic group (P < 0.001). **d** Scores of immunochemistry staining of CALML3 in tumours tissues of 100 patients with pulmonary metastasis and 170 patients without pulmonary metastasis. The horizontal lines in the box plot (minimum to maximum) represent the median and the interquartile ranges. Error bars represent mean ± SD. IHC scores were compared by Student's t-test, **P < 0.01. **e**, **f** Kaplan–Meier analysis showed that intratumoural CALML3 expression was a protective factor related to overall survival (P < 0.001) (**e**) and relapse-free survival (P < 0.001) (**f**). P-value is determined by log-rank test

CALML3 expression was significantly correlated with OS (P < 0.001) and RFS (P < 0.001); and that death and relapse were much more likely to occur in CALML3-negative patients than in CALML3-positive ones. The 1-, 3- and 5-year OS rates were 87.3%, 64.4% and 59.8%, respectively, in the intratumoural CALML3-positive group and 79.0%, 38.4% and 7.9%, respectively, in the intratumoural CALML3-negative group (P < 0.001). The 1-, 3- and 5-year RFS were 71.2%, 49.0% and 49.0%, respectively, in the positive group and 44.8%, 12.9% and 9.2%, respectively, in the negative group (P < 0.001) (Fig. 6e, f). These results suggested that CALML3 was a prognostic indicator of postoperative pulmonary metastasis and recurrence.

## Discussion

One major cause of high mortality of HCC is its high rate of metastasis. Thus, it is important to identify predictive biomarkers of the pre-metastatic state and understand the molecular pathology of metastasis initiation for early diagnosis or early prevention. To address this, we developed a prediction model of pulmonary metastasis, based on DNBs[11–13,15–17]. Compared with traditional molecular biomarkers detecting the metastatic state by differential expression of molecules in a static manner, DNB showed superiority in identifying the critical or pre-metastatic

state (a tipping point just before the dramatic transition[11,14–17] to a metastatic state) during disease progression via interactions between molecules (differential networks) in a dynamic manner. Then we determined the corresponding functional network of the DNB, which can signal the imminent transition. Another advantage of DNB method is that it is unnecessary to infer the unknown parameters, suggesting that it is a data-driven approach without requirements on parameters or even models.

Here, by analysing time-series transcriptomic data based on DNB method, we identified the tipping point of metastasis initiation (the third week after orthotopic implantation) using HCCLM3-RFP nude mice xenograft model, which was consistent with the CTC analysis by flow cytometry (Fig. 2e, g). Moreover, we found that CALML3 was one of DNB members and played an important role in metastasis initiation.

Human *CALML3* gene was located on chromosome 10 (10p15.1), encoding a 138-amino-acid residue calcium sensor protein, similar in structure to calmodulin (CaM)[39]. CaM was a single-chain protein composed of 148–152 conserved amino-acid residues[40]; it was a multifunctional protein that could form a $Ca^{2+}$–CaM complex to regulate signal transduction and metabolic activities related to calcium ion in eukaryotic cells[41]. However, there were few studies on physiological functions of CALML3. In previous work, CALML3, as a calmodulin-like protein, could bind

competitively with IQ protein on the neck of Myosin-10, which played an important role in cell migration[42]. There were other reports that CALML3 was highly expressed in normal differentiation of epithelial tissues, such as breast, thyroid, prostate, kidney and skin, and also significantly downregulated in breast tumours, oral cancers and skin cancers[39,43–45]. However, little is known about the role of CALML3 in HCC metastasis initiation.

Loss-of-function for tumour suppressor genes and gain-of-function for oncogenes are closely related to cancer initiation and progression. To further explore the role of genes involved in cancer, new genetic approaches are broadly used. Recent years, the RNA-guided clustered regularly interspaced short palindrome repeats-associated nuclease Cas9 (CRISPR-Cas9) provides an effective means of introducing multiplexable targeted loss-of-function mutations at specific sites in the genome[46]. Tet-on system that allows activation of gene expression by doxycycline in a time-dependent manner was used in gain-of-function assay[47]. In this study, to better elucidate the role of CALML3 as a tumour suppressor in HCC metastasis, we performed endogenous genetic tests on gain-of-function and loss-of-function, respectively. Gain-of-function model was established using an inducible system, which had a time-dependent increase effect of CALML3 expression in HCCLM3 cells with high metastatic capacity and low CALML3 expression. Loss-of-function model was conducted by CRISPR-Cas9 in MHCC97L cells with relatively lower metastatic capacity and higher CALML3 expression than HCCLM3 cells. To further verify the role of CALML3 in HCC metastasis, we also conducted CALML3 overexpression and knockdown models by lentivirus-mediated expression and RNA interference in HCCLM3 cells and HepG2 cells. We found that CALML3, as a suppressor, could significantly inhibit HCC carcinogenesis and metastasis initiation in vitro and in vivo (Fig. 4, Supplementary Fig. 4). Moreover, we revealed the functional role of CALML3 at a network level in the initiation of metastasis. According to a knowledge-based molecular network, we identified the genes and biological functions that were directly affected by CALML3 during metastasis initiation (Fig. 5a, b). Moreover, by PCR array, we found that typical metastasis-related genes were regulated by CALML3 in a cascade (Fig. 5c). These results demonstrated that CALML3, as a tumour suppressor gene, played an important role in protecting against pulmonary metastasis by regulating multiple signalling pathways.

Furthermore, the expression level of CALML3 may indicate the degree of tumour malignancy for HCC, and especially pulmonary metastasis initiation. Generally, surgical resection is one of the most effective treatments for HCC, but the high rate of metastasis and recurrence after resection restricts the long-term effect of surgery. Recently, clinicians have largely used clinical pathological factors (e.g., tumour encapsulation, tumour number, tumour size, vascular invasion, Edmondson grade and BCLC stage) to evaluate the risk of metastasis and recurrence after resection. Many molecular markers such as AFP, VEGF, CK19, CD133, osteopontin, and epithelial–mesenchymal transition (EMT) markers (e.g., vimentin, Twist, Snail, Slug and E-cadherin) are also considered in the study[48–51]. In this work, we compared CALML3 with existing HCC metastasis-related EMT markers, i.e., VEGF and vimentin by evaluating IHC. Receiver operating characteristic (ROC) analysis was performed to evaluate the performance of CALML3, VEGF and vimentin in distinguishing patients with HCC recurrence from patients without recurrence. CALML3 AUC (0.722, 95% CI: 0.657–0.787) was higher than VEGF AUC (0.641, 95% CI: 0.572–0.711) and vimentin AUC (0.671, 95% CI: 0.604–0.739). CALML3 had a sensitivity of 70.8% and a specificity of 73.6%, higher than those of VEGF (67.8% for sensitivity and 60.4% for specificity) and vimentin (65.5% for sensitivity and 68.7% for specificity) (Supplementary Fig. 5 and Supplementary

Table 5). The result of ROC analysis showed that compared to VEGF and vimentin, CALML3 has a better performance of distinguishing patients with HCC recurrence from patients without recurrence. But as a suppressor gene and an indicator of HCC development, CALML3 expression was significantly reduced in HCC tumour tissues and thus was relatively harder to be detected than oncogenes with high expression. That is the disadvantage of CALML3 from the traditional viewpoint of clinical application. However, with the development of novel tools and detection techniques in future, biomarkers such as regressed genes may become easier to be detected clinically. Thus, based on IHC analyses in tumour tissue slices from patients, we showed that CALML3 was a predictor of HCC metastasis, to guide the clinical diagnosis and early treatment of HCC patients.

In conclusion, our results suggested a new way to identify the tipping point of HCC pulmonary metastasis with its DNBs, and provided biological insights into the molecular pathology of this progression from the perspectives of dynamics and network. We also proposed CALML3 as a predictive biomarker as an early-warning indicator of the initiation of metastasis for clinical application.

## Methods

**Antibodies.** CALML3 (ab155130) and beta actin (ab8226) were from Abcam (Cambridge, MA). Secondary antibodies conjugated with HRP were from Jackson ImmunoResearch Laboratories (West Grove, PA).

**Cell culture.** The four human hepatoma cell lines (PLC/PRF/5, HepG2, Bel-7402 and SMMC-7721 cells) and an immortalised human hepatocyte liver cell (THLE-3) were obtained from the Type Culture Collection of the Chinese Academy of Sciences (Shanghai, China). MHCC97L, MHCC97H and HCCLM3 cells were obtained from the Liver Cancer Institute of Zhongshan Hospital, Fudan University. All the cell lines passed the conventional tests of cell line quality control methods (e.g., morphology, isoenzymes, mycoplasma). The cell lines have passed the test of DNA profiling (STR) between 2012 and 2014. Hepatoma cells were maintained in high-glucose Dulbecco's modified Eagle's medium (DMEM, GIBCO) supplemented with 10% fetal bovine serum (Hyclone, Utah, USA). THLE-3 was maintained in BEGM from Lonza/Clonetics Corporation, Walkersville, MD 21793 (BEGM Bullet Kit; CC3170) plus 10% fetal bovine serum. Cells were maintained at 37 °C in a humidified incubator under 5% $CO_2$.

**Tumour-bearing and orthotopic transplantation mouse model.** Ethical approval was obtained from the Research Ethics Committee of Zhongshan Hospital. Four-week-old male athymic BALB/c nu/nu mice were obtained from Shanghai Slac Laboratory Animal Co., Ltd., China. Mice were housed in laminar-flow cabinets under specific pathogen-free conditions. Mice were kept for 5–7 days as an adaptation period before being used in experiments. To assess the tumourigenicity of cells, tumour-bearing mouse model were conducted. A suspension of $1 \times 10^7$ cells in 0.2 ml phosphate buffer solution (PBS) were injected into the right upper flank of mice. Tumour sizes were recorded twice a week. Mice were sacrificed at 4 or 5 weeks post-injection; tumours were excised and weighed. Tumour volume was calculated by the formula: $0.5 \times L \times W^2$ ($L$ = length of tumour; $W$ = width of tumour). To examine the metastatic potential of cells, orthotopic liver tumour implantation model were conducted. A suspension of $1 \times 10^7$ cells in 0.2 ml PBS were injected into the right upper flank of mice. When the subcutaneous tumours reached 1 cm in diameter, 3 or 4 weeks later, they were removed, cut into pieces, of about $2 \times 2 \times 2$ mm$^3$, and implanted into the livers of another nude mice, as described previously[17]. To study of the whole-genome expression, 20 nude mice that underwent orthotopic tumour implantation, were divided into four groups. After the orthotopic tumour implantation, one group of mice were sacrificed at the second, third, fourth and fifth weeks for autopsy. For study the effect of inducible CALML3 expression by Tet-on technology on HCC metastasis, mice were received doxycycline injection (20 mg/kg) daily. Mice were euthanized at week 7 after tumour implantation. Orthotopic tumours and spontaneous metastasis to the lung were imaged (stereomicroscope: Leica MZ6; illumination: Leica L5 FL; C-mount: 0.63/1.25; CCD: DFC 300FX). Tumours and lung nodules were analysed histologically. Consecutive tissue sections (110–135 sections) were made for each block of the lung, and stained with haematoxylin-eosin.

**Whole-genome expression profile.** To assess whole-genome expression, we individually extracted total RNAs from liver tumours of orthotopic xenograft mice using TRIZOL Reagent (Life technologies) and checked for a RIN number to inspect RNA integrity by an Agilent Bioanalyzer 2100 (Agilent technologies). The qualified total RNAs were further purified by RNeasy micro kit (QIAGEN) and

RNase-Free DNase Set (QIAGEN). Then, total RNAs were amplified, labelled and purified by using GeneChip 3′IVT Express Kit to obtain biotin labelled cRNA (Affymetrix). Array hybridisation and wash were performed with constant rotation on the PrimeView Human Gene Expression Assay (Affymetrix). Microarrays were scanned by GeneChip Scanner 3000 (Affymetrix) and Command Console Software 3.1 (Affymetrix) with default settings. Raw data were normalized by robust multiarray analysis (RMA) algorithm, Gene Spring Software 11.0 (Agilent technologies). All microarray data were deposited in the Gene Expression Omnibus database (http://www.ncbi.nlm.nih.gov/geo/) Accession Number GSE94016.

To identify DEGs, we compared gene expression intensities among samples at two different time points using Welch's $t$-test with two-tailed $P$-value $< 0.05$, which was adjusted by FDR for multiple testing.

**Flow cytometry analysis**. Mice peripheral blood mononuclear cells (PBMC) were isolated from fresh blood by density gradient centrifugation through Ficoll-Hypaque (Sigma, St. Louis, MO). Then, single-cell suspensions were assessed on a FACS Aria II flow cytometer (BD Biosciences, CA, United States) and data were analysed using the Flowjo software (version 7.6.1). HCCLM3-RFP cells (with red fluorescence) acted as a positive control and wild-type HCCLM3 as a negative control.

**Tet-on induced CALMAL3 expression in HCCLM3 cells**. The human CALMAL3 sequence was cloned into Lenti-X Tet-one System expression vector (Clontech, Moutain View, CA). The recombinant lentivirus tet-CALMAL3 and the negative control lentivirus (Hanyin Co. Shanghai, China) were prepared and titered to $10^9$ TU/ml (transfection unit). To obtain cell lines expressing CALMAL3 only at the time when doxycycline existing, HCC cells were infected with lentiviruses expressing Tet-CALMAL3. At 48 h post-infection, 1 µg/ml of puromycin was supplemented in medium for selection.

**CRISPR/Cas9-mediated depletion of CALML3 in MHCC97L cells**. The 20 nt target DNA sequence guide RNA #1 (5′- TGCATCACCACCCGCGAGCT -3′) and guide RNA #2 (5′- GAGATCGACCGGGACGGCAA -3′) were selected for generating single guide RNA (sgRNA) for depleting CALML3 using the CRISPR design tool (http://crispr.mit.edu/). The targeted gRNA expression oligos were introduced into the PHY-701 vector (Hanyin Co. Shanghai, China)), which was modified according to letiCRISPRv2 (Addgene plasmid #52961). PHY-701-CALML3 sgRNA plasmids with pMD2.G and psPAX2 (Addgene plasmid #12259 and #12260) were co-transfected into 293T cell to make lentivirus. The recombinant sgRNA expression lentivirus were prepared and titered to $5 \times 10^8$ TU/ml (transfection unit). MHCC97L cells grown to 50% confluence in 6-well plate were added sgRNA lentivirus and selected with puromycin (1 µg/ml) for 48 h. Individual colonies were selected and genotyped by genomic DNA sequencing.

**Lentivirus-mediated CALML3 expression or knockdown clones**. CALML3 overexpression and knockdown stable clones were established via lentiviral-mediated expression and RNA interference, respectively. A panel of lentiviral particles with CALML3 target knockdown sequences (shRNA1: 5′-GCA-GAGCTGACCTTAGGAC-3′; shRNA2: 5′-GCGGGACATGATGAGTGAGAT-3′; shRNA3: 5′-GGAGAAGCAGAGCTGACCTTA-3′; shRNA4: 5′-GACAGGT-GAACTACGAGGAGT-3′; shRNA5: 5′-CGGAGACGGACAGGTGAACTA-3′), CALML3 cDNA fragment, and negative control particles (5′-GTAGCGCGGTG-TATTATAC-3′) were purchased from Hanyin Biotechnology Company (Shanghai, China). HCCLM3 and HepG2 cells were seeded in 6-well plates at a density of $5 \times 10^5$ cells per well in high-glucose DMEM supplemented with 10% fetal bovine serum and allowed to reach 90% confluence on the day of infection. Cells were transfected at a MOI of 30 and cell culture supernatants were replaced with fresh medium after a 4 h incubation with lentivirus. Then, 1 µg/ml puromycin was used to selected stable cell lines after 48 h of infection. The efficiency of overexpression or knockdown was examined by western blot analysis.

**Western blot**. The cells were washed twice with ice-cold PBS and then lysed with a modified radio-immunoprecipitation assay buffer (50 mM Tris-HCl, pH 7.4, 1% v/v NP-40, 0.25% v/v sodium deoxycholate, 150 mM NaCl, 1 mM EDTA, 1 mM PMSF, 1 mg/ml of protease inhibitors (leupeptin and pepstatin), 1 mM $Na_3VO_4$, and 1 mM NaF). Lysates were cleared by centrifugation and denatured by boiling in Laemmli buffer. Equal amounts of protein samples were loaded per well and separated on sodium dodecyl sulphate-polyacrylamide gels, and then transferred electrophoretically onto polyvinylidene fluoride membranes. Following blocking with 5% non-fat milk at room temperature for 1 hour, membranes were incubated with primary antibodies (anti-CALML3, 1:1000 dilution, Abcam, Cambridge, MA, USA) at 4 °C overnight and then incubated with horseradish peroxidase-conjugated secondary antibodies (1:5000 dilution) for 1 h at room temperature. Specific immune complexes were detected with chemiluminescence reagents (Western Blotting Chemiluminescence Reagent Plus, LifeScience).

**Cell proliferation**. Cell proliferation was examined using Cell Counting Kit-8 (CCK-8, Dojindo Co., Kumamoto, Japan) or Cell-IQ (Chip-man, Tampere, Finland). For CCK-8, Cell proliferation was measured according to the manufacturer's instructions. Cells were incubated with CCK-8 for 1 h with five multiples. Cell

proliferation rate was assessed by measuring the absorbance at 450 nm with the Universal Micro-plate Reader. For Cell-IQ, about $1 \times 10^3$ cells per well were plated in 24-well plates and incubated for 24 h. Then, the plates were transferred to Cell-IQ incubator with the cell-secure lid, and all-in-focus imaging recorded cells at 30-min intervals for 48 h. Analysis was carried out with a public-domain imaging software (McMaster Biophotonics Facility, Hamilton, ON, Canada), using the Manual Tracking plug-in created by Fabrice Cordelières (Institut Curie, Orsay, France). All assays were performed in triplicate and repeated three times.

**Wound healing scratch assay**. The migration of cells was evaluated with a wound healing scratch assay. Cells were seeded in 6-well plates ($1 \times 10^6$/well) in culture medium and cultured until confluent. Then the cells were treated with or without doxycycline (50 ng/ml). Then, an artificial scratch wound was drawn at the centre of the well and photographed. After 24 h of culture, the cell scratch wound was photographed again, and the migration distance was determined by the ratio of healing width at 24 h vs. the wound width at 0 hour. Each assay was carried out in triplicate and repeated three times.

**Transwell invasion assay**. The cell invasion assay was performed using 24-well matrigel invasion chambers with 8-µm pore inserts (BD Biosciences). HCCLM3 ($1 \times 10^5$) cells and MHCC97L ($5 \times 10^5$)cells were seeded into the upper chambers, which were filled with 2% FBS-containing culture medium, with or without doxycycline (50 ng/ml). The lower chambers were filed with 10% FBS-containing culture medium complete media. After 48 h of incubation, cells that migrated through the matrigel and adhered to the lower chamber were fixed in 4% paraformaldehyde for 20 min, and stained with crystal violet. For quantification, five fields per filter were counted under a microscope. Each invasion assay was carried out in triplicate and repeated three times.

**RT$^2$ profiler PCR array**. To investigate the molecular mechanism of CALML3 in HCC metastasis, we used the Human Tumour Metastasis RT$^2$ Profiler PCR Array, which is intended to represent 84 genes known to be involved in metastasis (KangChen Bio-tech Company, Shanghai, China). Genes selected for this array included several classes of proteins involved in cell adhesion, ECM components, cell cycle, cell growth and proliferation, apoptosis, transcription factors, regulators and other genes related to tumour metastasis. RNA isolation, DNase treatment and RNA clean-up were performed according to the manufacturer's protocol (Qiagen, Hilden, Germany). The isolated RNA was reverse transcribed into cDNA using the RT$^2$ First Strand Kit (Invitrogen). PCR was performed using the RT$^2$ SYBR Green qPCR Master Mix (Invitrogen) on an ABI PRISM7900 instrument (Applied Biosystems, Foster City, CA). Data normalization was based on correcting all Ct values for the average Ct values of several housekeeping genes present on the array. Each assay was conducted in triplicate.

**Patients and clinical follow-up**. From January 2009 to December 2011, curative resections were performed for HCC on 3544 patients, defined as macroscopically complete removal of the tumour, in the Liver Cancer Institute, Zhongshan Hospital, Fudan University. Of them, 100 cases were selected as the metastasis group, with pulmonary metastasis, and 170 cases as the control group, without pulmonary metastasis, from a surgical specimen database during the follow-up. None of the patients had received any preoperative anti-cancer treatment. Preoperative liver function was classified as Child-Pugh class A. The staging of HCC was determined according to the BCLC staging system. Tumour differentiation was graded according to the Edmondson grading system. Detailed clinicopathological features are provided in Supplementary Table 1. All subjects were contacted every 3 months during the first postoperative year, and at least 6 months afterward for survival and recurrence inquiries until death, contact failure, or the end of the investigation. A diagnosis of pulmonary metastasis was determined based on a typical imaging appearance in chest CT scans. The study was approved by the Zhongshan Hospital Research Ethics Committee. Informed consent was obtained according to the committee's regulations.

**Tissue microarray and immunohistochemistry**. Immunohistochemical staining for molecules of interest was performed on tissue microarrays (TMAs) made of formalin-fixed, paraffin wax-embedded tumour resection specimens. Two cores were taken from each representative tumour tissue (intratumoural) and from liver tissue adjacent to the tumour within a distance of 10 mm (peritumoural) to construct TMA slides (in collaboration with Shanghai Biochip Company Ltd., Shanghai, China). Then, two TMA sections with 270 pairs of tumours and matched peritumoural samples were constructed. Paraffin wax-embedded sections were deparaffinised in xylene and rehydrated in a decreasing ethanol series, diluted in distilled water. Following antigen retrieval with 10 mM citrate buffer, TMA sections were incubated overnight at 4 °C with a CALML3 antibody (Abcam Inc., Cambridge, MA, USA). Following 30 min incubation with secondary antibody, sections were developed in 3,3′-diaminobenzidine solution under microscopic observation and counterstained with haematoxylin. The sections were observed under a light microscope for a histological review, to examine tumour microheterogeneity in terms of antigen distribution. Five randomised microscopic views of each section at ×100 and ×400 magnification were observed and scored. The semi-quantitative

scoring system was based on both staining intensity (0, negative; 1, weak; 2, intermediate; 3, strong) and the percentage of positive cells (0, 0% positive cells; 1, 1–25% positive cells; 2, 26–50% positive cells; 3, 51–75% positive cells; 4, 76–100% positive cells)[52]. The evaluation was carried out by two independent pathologists who were unaware of the patient outcomes. Images were acquired with a Pannoramic MIDI Slide scanner (3D HISTECH, Budapest, Hungary)

**DNB analysis**. Recently, many studies on complex diseases have suggested that the development and progression of diseases are not always slow and linearly changing phenomena but have occasional marked and nonlinear transitions[11–17] (Fig. 2c). Such a transition changes the state of the biological system qualitatively, and thus plays key roles in biological processes. Clearly, identifying the critical state or tipping point just before this transition is important for preventative measures. Thus, to identify the predictive biomarkers and understand the molecular mechanisms of metastasis initiation for early diagnosis and prevention, we developed a mathematical model, i.e., the DNB method[11,12,14,15,17], which can detect early-warning signals of pulmonary metastasis for primary tumour cells. Based on nonlinear dynamic theory from the measured data, we have theoretically and numerically shown[11,17] that if there is a dominant group of molecules or genes, i.e., DNBs (dynamic network biomarkers), satisfying the following three criteria from the observed data, then the system is near the critical state or tipping point:

(1)  Standard deviations ($SD_i$) of molecules in this dominant group are increased markedly,
(2)  Pearson correlation coefficients ($PCC_i$) of molecules (expression levels) in this dominant group are increased significantly and
(3)  Pearson correlation coefficients between molecules in this group and others ($PCC_o$) are decreased significantly.

The following quantification index (CI: criticality index) approximately considering the all three criteria can be used as the numerical signal of the DNB method:

$$CI = \sqrt{size}\,\frac{PCC_i}{PCC_o}SD_i$$

where size is the number of molecules in the dominant group or DNB, $SD_i$ is the average standard deviation of all molecules in the dominant group, $PCC_i$ is the average PCC of all molecule-pairs in the dominant group (absolute value), and $PCC_o$ is the average PCC of molecule-pairs between the dominant group and others (absolute value). In fact, the three conditions can be expressed approximately in the following sentence: from the measured data, the appearance of a group of genes (or proteins) with strongly collective fluctuations indicates an imminent critical transition (i.e., the system is near the critical state). Thus, the group members are the predictive/dynamic biomarkers for this critical (generally irreversible) transition. When CI reaches a peak or increases markedly during the measured periods, the biological system is at the critical period or tipping point (Fig. 2e). The DNB distinguishes not metastatic samples but pre-metastasis samples from non-metastasis samples using both molecular fluctuation information (i.e., dynamic information) and network information (i.e., correlation information among molecules)[11,17], in contrast to the traditional static biomarkers. A detailed algorithm is shown in Supplementary Fig. 1.

**Ranking scheme for DNB members**. We ranked the DNB genes according to the following criteria with four priorities.

(1)  Priority one (the first bar in Fig. 3a). Rank DNB genes for their importance at the network level. Hub genes are considered to play a central role in the molecular network (including DNB genes and DEGs) during metastasis initiation. A hub from DNB genes is a gene with a number of links (i.e., degree) that considerately exceeds the average in the network. Thus, we mapped both DNB genes and DEGs into the molecular network and then individually counted the total number of DEGs directly linked with each DNB gene, i.e., obtained the degree of each DNB gene. This criterion is represented as a percentage or ratio of DEGs to its neighbouring genes.
(2)  Priority two (the second bar in Fig. 3a). Rank DNB genes based on their functional roles at biological pathway level. We counted the number of metastasis-associated pathways involving each DNB gene. Thus, we ranked these DNB genes according to the total number of their involved KEGG-annotated pathways[27]. In this work, 163 pathways[27] were considered to be related to metastasis. These 163 pathways include most metabolism processes (e.g., carbohydrate, energy, lipid, amino acid), and all genetic information processing, cellular processes, and signalling pathways (Supplementary Data 3).
(3)  Priority three (the third bar in Fig. 3a). This criterion indicates if or not a DNB gene belongs to one of DEGs, i.e., 1 (yes) or 0 (no).
(4)  Priority four (the fourth bar in Fig. 3a). This criterion indicates if or not a DNB gene belongs to one of the six clusters, i.e., 1 (yes) or 0 (no). If yes, this criterion further indicates, which cluster this DNB gene belongs to. In other words, this criterion describes how each DNB gene is related to the role of suppressors or oncogenes at metastasis initiation. Specifically, we checked which DNB genes overlapped with DEGs in Clusters 1 to 6 (Fig. 2b).

According to the change in patterns at the gene expression level during W2–W4, we considered that the DNB genes in Clusters 1, 3 and 5 might act as (or be related to) suppressors, preventing from metastasis, because they were downregulated before the metastasis; whereas other DNB genes in Clusters 2, 4 and 6 might function as (or be related to) oncogenes, promoting metastasis, because they were upregulated before metastasis.

**Statistical analysis**. GraphPad Prism 5.0 software (GraphPad-Prism Software Inc., San Diego, CA) and SPSS 19.0 (Chicago, IL, USA) was used for statistical analyses. Cumulative survival time was calculated by the Kaplan–Meier method and analysed with the log-rank test. Univariate and multivariate analyses were based on the Cox proportional hazards regression model. ROC curve analysis was used to determine the predictive value of the parameters, and the differences in the area under the curve (AUC) were detected by SPSS. The $\chi^2$ test and Student's $t$-test were used for comparison between groups. All data were presented as mean ± SD from three independent experiments. $P$-value < 0.05 was considered statistically significant.

**Functional analysis**. To uncover potential HCC-associated biological functions regulated by the identified genes or proteins, we mapped the proteins into known molecular sets with functional annotations. Here, KEGG[53] was used for canonical pathway detection. Then, we estimated enrichment significance of specific proteins in each biological process or pathway based on a hypergeometric test. Thus, significantly enriched functions were chosen by the corresponding $P$-value being < 0.05 after FDR correction.

**Accession numbers**. All data have been uploaded to GSE94016.

**Data availability**. The data that support this study are available within the article and its Supplementary Information files or available from the authors upon request.

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

## Acknowledgements

We are thankful to National Key R&D Program of China (No. 2017YFA0505500), Strategic Priority Research Program of the Chinese Academy of Sciences (CAS) (No. XDB13040700), the National Program on Key Basic Research Project (No. 2014CB910504), the National Natural Science Foundation of China (NSFC) (No. 91439103, 91529303, 81471047, 81572395, 31771476), and the Shanghai Science and Technology Commission (grant numbers 14XD1401100).

## Author contributions

L.C., J.X., B.Y., M.L. designed and conceived this project. M.L., L.C. developed methodology; B.Y., W.T. generated data and performed experiments; M.L., W.L. analysed and interpreted data; M.L., B.Y., W.T., L.C., J.X. wrote the manuscript. S.Z. contributed to the design of biological experiments. All authors contributed to and approved the manuscript.
