## [Peer Review File · Nature Communications]

Reviewers' comments:

Reviewer #1 Expert in molecular biology and DNB:

The paper by Yang et al. describes a somewhat novel approach (DNB theory) for identifying novel genes important for the metastatic potential of hepatocellular carcinomas.

The main novelty as compared to previous publications by the main authors is the actual testing of one key prediction of the approach: that is the tumor suppressor role for the CALML3 gene.

Although this is an important paper, as it stands it is clearly not up to the level for the proposed journal, mainly because too many (essential) "details" are left unaddressed.

1. It is really annoying that the actual transcriptomic data (so called "whole genome profile") on which the whole study is based is not explained at all. Is this microarray? RNASeq? How are raw data processed? This has to be clearly explained.
2. Overall, the manuscript is kind of sloppy. For example, "Pearson's correlation correlation" in sup figure 1 or "to overexpress CALML3 in HepG2 cells, with low metastatic potential, and to decrease expression in HCCLM3 cells, with high metastatic potential " (lanes 287-288): this is the reverse! Line 368: "CALML3 expression and so on..."
3. The "detailed" algorithm of the DNB method is not detailed enough. What tests are used for "significantly high deviations"? What tests are used to test for deciding whether a correlation is "high compared to control"? What would be the control here?
4. Lane 218: the authors write: "This result is consistent with previous results analysing the dynamics of gene expression" What are the (unreferenced) "previous results " here?
5. How many genes are finally being considered as part of the DNB? A list of those genes could be useful (if not too long)?
6. The procedure for choosing CALML3 among the DNB genes is really not properly described. Are not DNB genes a subset of DEGs? If so (and this seems to be the case from Figure 3a) why is it being taken into account? Figure 3a is confusing. What percent is represented among the y axis? This should be made clear in the legend. Why is the PIK3R5 gene not considered as a better candidate than the CALML3 gene?
7. A more deep question is: the author assume that CALML3 is kind of the driver ("regulating these reversing DEGs at a molecular network level."). Two questions are immediately obvious: how (i.e. by what molecular mechanism?) and how direct? (i.e. not comparing two established cell lines but a time dependent increase/decrease using an inducible system).
8. In the discussion, the authors write that their approach is "model free". This I think is a misleading assumption: they assume the existence of a dynamical system in which genes-to-genes interaction do exist and can be captured using a simple correlation coefficient. This is a strong assumption which characterizes a model in my view (although I fully agree that they do not try to infer the parameter values of such a dynamical system, they assume its existence).

In the end a much more precise and rigorous description of how the DNB is isolated, and how the CALML3 gene is chosen is mandatory.

Reviewer #2 Expert in HCC and systems biology:

The authors reported a comprehensive study of HCC pulmonary metastatic cell lines, PDX mouse model and clinical samples. They applied a mathematical model (DNB) to identify CALML3 as a key DNB and a HCC/metastasis suppressor that could trigger metastasis initiation and also predict patient survival rates. Overall this is a nice multi-interdisciplinary and integrative analysis, and the dataset is also valuable. But, since the mechanistic study is lacking, many claims are not really proved and hence must be restated appropriately:

- (1) Proving a tumor suppressor requires endogenous genetic tests on "loss of function" and "gain

of function". Available data could only support a putative tumor suppressor at best. Similarly for metastatic suppression.

(2) All analyses are correlative results that cannot give direct and causative or mechanistic conclusions. There is NO proof that CALML3 (or any other DNB) "triggers pulmonary metastases" (in fact, it cannot be the "trigger" as explained below), hence several claims in the manuscript (including the title) are not substantiated by current data.

(3) Down-regulation of CALML3 may be related to EMT, but unlike Snail, from its known function, CALML3 is more likely to be a consequence than a cause ("driver" or "trigger"). Indeed in signaling, a good cancer marker needs not at all to be a "driver" or "trigger"; on the contrary, a "trigger" is often difficult to detect but the consequence is usually much easier to detect and hence can serve as a better "marker".

(4) As a biomarker, a regressed gene may not be a GOOD marker clinically than an amplified gene from the point-of-view of easy detection. The authors should at least compare CALML3 with existing biomarkers and discuss its advantages and disadvantages. When claiming "CALML3 is an independent predictor of HCC metastasis and recurrence", the authors should show if it is truly independent from other EMT markers, e.g. VEGF markers?

There are several other similar "overstatements" or "inaccurate" minor statements that need to be revised: e.g.

(5) in "Discussion", "..DNB.. " is "model-free"..."without requirements on parameters or even models" which contradict with "we developed a new mathematical model..." and the three criteria serving as the parameters to be estimated from the data.

(6) "In conclusion, our results suggest a new way to identify the tipping point or critical state with its dynamic biomarkers", this contradicts with introduction where it says DNB was already published several years ago and has been applied to several other systems, including cancer studies.

If the authors could revise and address these questions, it would be acceptable for publication in NM.

Response to the Comments of Reviewer #1

Reviewer's Comment 1:

It is really annoying that the actual transcriptomic data (so called "whole genome profile") on which the whole study is based is not explained at all. Is this microarray? RNASeq? How are raw data processed? This has to be clearly explained.

Authors' Response:

Thanks for your comment. To address this question, we added the detailed description of transcriptomic data in Methods of the revised manuscript, as follows.

"Whole-genome expression profile

To assess whole-genome expression, we individually extracted total RNAs from liver tumours of orthotopic xenograft mice using TRIZOL Reagent (Life technologies) and checked for a RIN number to inspect RNA integrity by an Agilent Bioanalyzer 2100 (Agilent technologies). The qualified total RNAs were further purified by RNeasy micro kit (QIAGEN) and RNase-Free DNase Set (QIAGEN). Then, total RNAs were amplified, labelled and purified by using GeneChip 3'IVT Express Kit to obtain biotin labelled cRNA (Affymetrix). Array hybridization and wash were performed with constant rotation on the PrimeView Human Gene Expression Assay (Affymetrix). Microarrays were scanned by GeneChip Scanner 3000 (Affymetrix) and Command Console Software 3.1 (Affymetrix) with default settings. Raw data were normalized by robust multiarray analysis (RMA) algorithm, Gene Spring Software 11.0 (Agilent technologies). All microarray data were deposited in the Gene Expression Omnibus database (<http://www.ncbi.nlm.nih.gov/geo/>) Accession Number GSE94016.

To identify differentially expressed genes, we compared gene expression intensities among samples at two different time points using Welch's t-test with two-tailed p value < 0.05 which was adjusted by False Discovery Rate (FDR) for multiple testing."

Reviewer's Comment 2:

Overall, the manuscript is kind of sloppy. For example, "Pearson's correlation correlation" in Supplementary Fig. 1 or "to overexpress CALML3 in HepG2 cells, with low metastatic potential, and to decrease expression in HCCLM3 cells, with high metastatic potential "(lanes 287-288): this is the reverse! Line 368: "CALML3 expression and so on...".

Authors' Response:

We appreciate that you checked our paper carefully and helped us to improve the quality of this paper. According to your comment, we double-checked and carefully revised our paper. In particular, we made the following revisions in the revised manuscripts:

1. "Pearson's correlation correlation" was replaced with "Pearson correlation coefficient" in Supplementary Fig. 1. Meanwhile, to address your next question, we added the description of DNB algorithm given in Supplementary Fig. 1, which is explained in the response of the next comment.

2. We made inappropriate descriptions in the previous version of the manuscript, which have been corrected in the revised manuscript as follows.

"In addition, to further verify the role of CALML3 as a tumour suppressor, we overexpressed CALML3 in HCCLM3 cells with high metastatic potential and decreased CALML3 expression in HepG2 cells with low metastatic potential by lentiviral-mediated expression and RNA interference, respectively (Supplementary Fig. 4a)."

Supplementary Figure 4. CALML3 could suppress HCC metastasis *in vitro* and *in vivo*. (a) The efficiency of CALML3 overexpression in HCCLM3 cells and knockdown in HepG2 cells by lentivirus mediated expression and RNA interference was tested by Western blot. Of four different clones of CALML3 knockdown, sh4 produced the best knockdown efficiency and was chosen for further functional assays named shCALML3.

3. We re-described the result for easy reading, i.e., "In total, sixteen Clinical characteristics (e.g. tumour encapsulation, vascular invasion, tumour number, tumour size, BCLC stage, intratumoural CALML3 expression and so on) were analysed to identify factors associated with overall survival (OS) and relapse-free survival (RFS)."

Reviewer's Comment 3:

The "detailed" algorithm of the DNB method is not detailed enough. What tests are used for "significantly

high deviations”? What tests are used to test for deciding whether a correlation is “high compared to control”? What would be the control here?

Authors’ Response:

According to your suggestion, we added the details of DNB algorithm in Supplementary Fig. 1.

1. What tests are used for “significantly high deviations”?

For deviation test, the criterion is for one gene more than 2 times of the ratio (SD_d) of standard deviation at t to at t_0 .

2. What tests are used to test for deciding whether a correlation is “high compared to control”? What would be the control here?

We considered gene expression data of samples at the second week after orthotopic implantation (i.e. $t=t_0=0$) as control (see Supplementary Fig. 1). Here, “compared to control” means that all three criteria (i.e., SD_d , PCC_{id} , PCC_{od}) are ratios of the corresponding parameters at each time point t to at t_0 as indicated in the algorithm (Supplementary Fig. 1). For all PCC values, we normalized them by Fisher z-transformation and the threshold of clustering is the quantile $Z_{0.99}$ of this normal transformation.

In light of your comments, we added Supplementary Fig. 1 and the related descriptions in the revised manuscript.

Figure S1
Supplementary Figure 1. The detailed algorithm of DNB method. A flowchart describes the detailed steps to determine DNB members based on our mathematical method.

Reviewer's Comment 4:

Lane 218: the authors write: "This result is consistent with previous results analysing the dynamics of gene expression" What are the (unreferenced) "previous results" here?

Authors' Response:

Here, we revised this following sentence with clear illustration in light of the comment.

"This result was consistent with the morphological alterations in orthotopic xenograft mice (Fig. 1b, c) and our assumption from the dynamics of gene expression (Fig. 2b)."

Reviewer's Comment 5:

How many genes are finally being considered as part of the DNB? A list of those genes could be useful (if not too long)?

Authors' Response:

The DNB including 334 genes was detected by our method and it is too many to be listed in the main text. Thus, we added the following description and also Supplementary Table 2 in the revised manuscript.

"Meanwhile, we obtained the corresponding leading group as DNB which was composed of 334 genes or proteins (Supplementary Table 2)."

Reviewer's Comment 6:

The procedure for choosing CALML3 among the DNB genes is really not properly described. Are not DNB genes a subset of DEGs? If so (and this seems to be the case from Figure 3a) why is it being taken into account? Figure 3a is confusing. What percent is represented among the y axis? This should be made clear in the legend. Why is the PIK3R5 gene not considered as a better candidate that the CALML3 gene?

Authors' Response:

1. The procedure for choosing CALML3 among the DNB genes is really not properly described.

There are 334 genes in the DNB group. It is really a difficult task to measure the importance of a gene in a

molecular network due to so many involved factors, and there may be many different ways to do it. To validate the important role of DNB, in this manuscript we considered the following criteria with the priority to rank these DNB genes for conducting biological experiment and added the detailed description in Methods.

“Ranking scheme for DNB members

We ranked the DNB genes according to the following criteria with four priorities.

(1) Priority one (the first bar in Fig. 3a). Rank DNB genes for their importance at the network level. Hub genes are considered to play a central role in the molecular network (including DNB genes and DEGs) during metastasis initiation. A hub from DNB genes is a gene with a number of links (i.e. degree) that considerably exceeds the average in the network. Thus, we mapped both DNB genes and DEGs into the molecular network and then individually counted the total number of DEGs directly linked with each DNB gene, i.e., obtained the degree of each DNB gene. This criterion is represented as a percentage or ratio of DEGs to its neighbouring genes.

(2) Priority two (the second bar in Fig.3a). Rank DNB genes based on their functional roles at biological pathway level. We counted the number of metastasis-associated pathways involving each DNB gene. Thus, we ranked these DNB genes according to the total number of their involved KEGG-annotated pathways. In this work, 163 pathways were considered to be related to metastasis. These 163 pathways include most metabolism processes (e.g., carbohydrate, energy, lipid, amino acid), and all genetic information processing, cellular processes, and signalling pathways (Supplementary Table 3).

(3) Priority three (the third bar in Fig.3a). This criterion indicates if or not a DNB gene belongs to one of DEGs, i.e., 1 (yes) or 0 (no).

(4) Priority four (the fourth bar in Fig.3a). This criterion indicates if or not a DNB gene belongs to one of the six clusters, i.e., 1 (yes) or 0 (no). If yes, this criterion further indicates which cluster this DNB gene belongs to. In other words, this criterion describes how each DNB gene is related to the role of suppressors or oncogenes at metastasis initiation. Specifically, we checked which DNB genes overlapped with DEGs in Clusters 1 to 6 (Fig. 2b). According to the change in patterns at the gene expression level during W2-W4, we considered that the DNB genes in Clusters 1, 3 and 5 might act as (or be related to) suppressors, preventing from metastasis, because they were downregulated before the metastasis; whereas other DNB genes in Clusters 2, 4 and 6 might function as (or be related to) oncogenes, promoting metastasis, because they were upregulated before metastasis.”

In Results, we also made the corresponding revision, as follows.

“To further understand the underlying roles of DNB genes in metastasis initiation for primary tumour cells,

we ranked comprehensively the DNB genes according to the criteria with four priorities, i.e. importance in the networks, importance in functional pathways, differential patterns, and dynamic patterns (see Methods), and selected CALML3 as the top one for further functional study (Fig. 3a, Supplementary Table 2). Certainly, based on the different considerations, there may be other ways to rank the genes.”

2. Are not DNB genes a subset of DEGs? If so (and this seems to be the case from Figure 3a) why is it being taken into account?

DNB genes were identified only based on the three criteria (i.e., drastic fluctuation at genes expression, increasing correlations between genes within DNB group, and decreasing correlations between genes within and without DNB group). Thus, there is no requirement on differential expressions for those DNB genes. In other words, a DNB gene may be a DEG or may be not a DEG gene. Hence, DNB genes are not necessary to be a subset of DEGs.

However, as mentioned above, we could estimate roughly what roles DNB genes play according to their changes in patterns and how DNB genes are associated with the DEGs at the gene expression level. Hence, we took this criterion into account. In light of the comment, we added the explanation in the revised manuscript.

3. What percent is represented among the y axis? This should be made clear in the legend.

As shown in Fig. 3a, 4 different bars represent criteria or attributes mentioned above to rank DNB genes. For the first bar, we used a percentage to represent how many DEGs were associated with the corresponding DNB gene (degree of each gene in the network) against the number of all neighboring genes for this DNB gene. Note that, in order to avoid bias, we only considered all neighboring genes (including non-DEGs) detected in this whole-genome expression profile. Thus, we re-calculated the corresponding data in Supplementary Table S2 as well as re-ranked the DNB genes and updated Fig. 3a. The second bar shows the total number of DNB-involved pathways. The third and the fourth bars represent individually if or not a DNB gene belongs to one of DEGs and one of the six clusters which changed during 2 to 4 weeks after orthotopic implantation. In addition, we also labelled the corresponding cluster of DEGs to clearly show how DNB gene expression changes during the critical period of metastasis. As mentioned above, there are different scales in this figure. Thus, to avoid the confusion, we removed the y axis and directly labelled information on the corresponding bars as follows (Fig. 3a). The further explanations were added in the revised manuscript.

Figure 3
Figure 3. CALML3 ranked as one of core DNB members and generation of CALML3 overexpression and knockout cells by Tet-on and CRISPR/Cas9 systems. (a) The diagram shows the DNB ranking, based on the various criteria with the four priorities, during the progression of metastasis. CALML3 was chosen as the top one for further functional tests. Note that the patterns of the change for a DNB gene at gene expression level were labelled as 1 or 0 (on the third and fourth bars). For DEG of W2-W4, C indicates a Cluster (e.g. C5 means that the gene belongs to Cluster 5).

4. Why is the PIK3R5 gene not considered as a better candidate than the CALML3 gene?

Indeed, DNB as a group, its members may play important roles during metastasis initiation. Here, we intended to choose not only important and but also novel one for further functional study based on our heuristic criteria. Undoubtedly, FGFR4, PIK3R5 and PDGFRA have been reported notably associated with carcinogenesis, which in turn, further validated the significance and effectiveness of our method to identify the important genes. However, there is less report about relations between CALML3 and cancer, especially in pulmonary metastases of hepatocellular carcinoma. Thus, we chose CALML3 to uncover its dysfunctions during pulmonary metastases of hepatocellular carcinoma. Other genes will be studied as our future work. In light of your comment, we added the related explanations in the revised manuscript.

Reviewer's Comment 7:

A more deep question is: the author assume that CALML3 is kind of the driver ("regulating these reversing DEGs at a molecular network level."). Two questions are immediately obvious: how (i.e. by what molecular mechanism?) and how direct? (i.e. not comparing two established cell lines but a time dependent increase/decrease using an inducible system).

Authors' Response:

As suggested by the reviewer, CALML3 may not be a kind of driver although DNB genes are considered to be strongly related to drivers. But rather than drivers, the main purpose of our study is to identify the tipping point of HCC metastasis by dynamic network biomarker that is able to signal the emergence of the disease transition. Thus, based on your comment as well as the suggestion from the reviewer #2, we

changed the description of CALML3 from “driver” to “indicator” of pulmonary metastases in the revised manuscript because it indicates the imminent transition of pulmonary metastases based on our data. In other words, we reconsidered CALML3 as an indicator of pulmonary metastases, which is also a suppressor based on the result of present study.

In addition, to further explore the role of CALML3 in HCC metastasis, we performed gain-of-function and loss-of-function studies to detect the effect of CALML3 on oncogenic behaviours of hepatoma cells including cell growth, migration and invasion. Considering that MHCC97L and HCCLM3 (metastatic potential: MHCC97L < HCCLM3) were established from the same parent HCC cell line and showed relative high and low CALML3 expression respectively, we established a doxycycline-inducible CALML3 overexpression system in the HCC cell line HCCLM3 and a CRISPR/Cas9 mediated knock out in the HCC cell line MHCC97L. Treatment with 50 ng/ml doxycycline induced a time-dependent expression of CALML3 protein in HCCLM3/CALML3 cells, detectable as early as 12 h (Fig. 3c). Two CALML3-targeting gRNAs sequences were designed to avoid non-specific effects of the CRISPR/Cas9 system. Confirmation of the genotype of two CALML3^{-/-} cell lines (CALML3^{-/-} #1 and CALML3^{-/-} #2) was testified by genomic DNA sequencing and Western blot (Supplementary Fig. 3 and Fig. 3d).

Cancer is characterized by sustaining proliferative signalling, evading growth suppressors, resisting cell death, enabling replicative immortality, inducing angiogenesis, and activating invasion and metastasis³⁸. First we examined cell proliferation ability of CALML3 overexpression or knockout cells and the corresponding controls. The result of cell proliferation assay showed that inducible CALML3 expression by Tet-on system significantly inhibited cell growth in HCCLM3 cells, while knockout of endogenous CALML3 by CRISPR/Cas9 technology could promote cell proliferation in MHCC97L cells (Fig. 4a). The results of cell migration and invasion assay showed that compared to the controls, CALML3 overexpression cells displayed markedly decreased ability of migration and invasion, however, CALML3 knockout had an opposite effect on metastasis of HCC cells (Fig. 4b-e). In accordance with assay *in vitro*, the result of xenografts models established by inducible CALML3 overexpression system in HCCLM3 cells verified that overexpression of CALML3 significantly diminished tumorigenic capacity and pulmonary metastasis. Inducible CALML3 expression group exhibited less weight and volume of tumours, as well as less pulmonary metastasis nodules and incidence than the control (Fig. 4f-i). In addition, to further verify

the role of CALML3 as a tumour suppressor, we overexpressed CALML3 in HCCLM3 cells with high metastatic potential and decreased CALML3 expression in HepG2 cells with low metastatic potential by lentiviral-mediated expression and RNA interference, respectively (Supplementary Fig. 4a). The results also showed that overexpression of CALML3 remarkably inhibited cell proliferation (Supplementary Fig. 4b), cell migration (Supplementary Fig. 4c,d), and cell invasion (Supplementary Fig. 4e,f), whereas downregulation of CALML3 had the opposite effect. We also created xenografts models of the control HCCLM3 and CALML3 overexpression HCCLM3 cells with fluorescent proteins as above (Fig. 1a); CALML3 overexpression significantly inhibited tumour growth and tumour pulmonary metastasis (Supplementary Fig. 4g,h). Together, the results of these functional assays *in vitro* and *in vivo*, suggested that CALML3 was a suppressor gene in HCC tumorigenesis and metastasis. In light of the comment, we added the results with new figures (Figs. 3-4, Supplementary Figs. 3-4) in the revised manuscript.

Figure 3

Figure 3. CALML3 ranked as one of core DNB members and generation of CALML3 overexpression and knockout cells by Tet-on and CRISPR/Cas9 systems. (a) The diagram shows the DNB ranking, based on the various criteria with the four priorities, during the progression of metastasis. CALML3 was chosen as the top one for further functional tests. Note that the patterns of the change for a DNB gene at gene expression level were labelled as 1 or 0 (on the third and fourth bars). For DEG of W2-W4, C indicates a Cluster (e.g. C5 means that the gene belongs to Cluster 5). (b) The protein

expression levels of CALML3 in normal human liver epithelial cell (THLE-3) and multiple cell lines with different metastatic potentials. The rectangles with red frames showed MHCC97L, MHCC97H and HCCLM3 (metastatic potential: MHCC97L < MHCC97H < HCCLM3), which were established from the same parent HCC cell line, MHCC97. (c) Establishment and characterization of a doxycycline-inducible CALML3 overexpression system. Upper panel: Schematic of Lenti-X Tet-One inducible expression system to express inducible CALML3. Dox, doxycycline; TRE, tetracycline responsive element. Lower panel: Western analysis of CALML3 in extracts derived from inducible CALML3 cells, either untreated or induced with doxycycline (50 ng/ml) at indicated time points. (d) Generation of CALML3-knockout MHCC97L cell lines (clones of CALML3^{-/-} #1 and CALML3^{-/-} #2 cell lines) using the CRISPR/Cas9 system. di: Schematic representation of the CALML3-targeting gRNA sequences. Arrows indicate primer positions. PAM, protospacer adjacent motif. dii: Two CALML3^{-/-} cell lines were established from MHCC97L cells. The deleted sequences in the CALML3^{-/-} #1 and CALML3^{-/-} #2 cell lines are presented. Individual colonies were selected and genotyped by genomic DNA sequencing. diii: Western analysis of CALML3 in extracts derived from control and two CALML3^{-/-} cell lines. β -actin was used as the loading control.

Figure S3

Supplementary Figure 3. Individual colonies (CALML3^{-/-} #1 and CALML3^{-/-} #2 cells) of CALML3 knockout by CRISPR/Cas9 were genotyped by genomic DNA sequencing. (a) CALML3^{-/-} #1 cells had deletions of 14bp in CALML3 genomic region. (b) CALML3^{-/-} #2 cells had deletions of 11bp in CALML3 genomic region. WT, wild type.

Figure 4

Figure 4. CALML3 acts as a HCC metastatic suppressor *in vitro* and *in vivo*. (a) CALML3 overexpression in HCCLM3 cells induced by doxycycline (50 ng/ml) for indicated time inhibited cell proliferation, while CALML3 knockout by CRISPR-Cas9 technology promoted cell proliferation. Two clones (CALML3^{-/-} #1 and CALML3^{-/-} #2) targeting CALML3 were designed for functional assays. (b) CALML3 overexpression suppressed cell migration (upper panel) while CALML3 knockout brought the opposite effect (lower panel). (c) Quantification of migration cells in the indicated groups in the wound healing assay. CALML3 overexpression showed obvious suppression of migration abilities in HCCLM3 cells (upper panel) while CALML3 knockout induced significant promotion of migration abilities in CALML3^{-/-} #1 and CALML3^{-/-} #2 MHCC97L cells (lower panel). (d) CALML3 overexpression suppressed cell invasion (upper panel) while CALML3 knockout brought the opposite effect (lower panel). (e)

Quantification of invasive cells in the indicated groups in the transwell invasion assay. (f) Tumours from mice implanted with HCCLM3 cells (the control and CALML3 overexpression) in tumour-bearing mouse model. (g) Comparison of tumour weight and size in tumour-bearing mouse model assay. CALML3 overexpression induced by doxycycline showed less tumour weight and significantly slowed tumour growth. (h) Lung tissues from mice implanted with HCCLM3 cells (control and CALML3 overexpression) orthotopic transplantation model were stained with hematoxylin-eosin. (i) Comparison of pulmonary metastasis in xenografts models. CALML3 overexpression induced by doxycycline showed less lung metastatic nodules and reduced incidence of lung metastasis. Dox, doxycycline. * $p < 0.05$, ** $p < 0.01$.

Figure S4

Supplementary Figure 4. CALML3 could suppress HCC metastasis *in vitro* and *in vivo*. (a) The efficiency of CALML3 overexpression in HCCLM3 cells and knockdown in HepG2 cells by lentivirus mediated expression and RNA interference was tested by Western blot. Of four different clones of CALML3 knockdown, sh4 produced the best knockdown efficiency and was chosen for further functional assays named shCALML3. (b) CALML3 overexpression suppressed cell proliferation in HCCLM3 cells while CALML3 knockdown promoted cell proliferation in HepG2 cells. (c) CALML3 overexpression suppressed cell migration (left panel) while CALML3 knockout brought the opposite effect (right panel). (d) Quantification of migration cells in the indicated groups in the wound healing assay. CALML3 overexpression showed obvious suppression of migration abilities in HCCLM3 cells while CALML3

knockdown induced significant promotion of migration abilities in HepG2 cells. (e) CALML3 overexpression suppressed cell invasion while CALML3 knockdown brought the opposite effect. (f) Quantification of invasive cells in the indicated groups in the transwell invasion assay. (g) Tumours from mice implanted with HCCLM3 cells (the control and CALML3 overexpression cells). CALML3 overexpression showed less tumour weight. (h) Lung tissues from mice implanted with HCCLM3 cells (the control and CALML3 overexpression) orthotopic transplantation model were examined by fluorescence microscopy (Left panel). CALML3 overexpression HCCLM3 cells showed less lung metastatic nodules (Middle panel) and reduced incidence of lung metastasis (Right panel). * $p < 0.05$, ** $p < 0.01$.

Reviewer's Comment 8:

In the discussion, the authors write that their approach is “model free”. This I think is a misleading assumption: they assume the existence of a dynamical system in which genes-to-genes interaction do exists and can be captured using a simple correlation coefficient. This is a strong assumption which characterizes a model in my view (although I fully agree that they do not try to infer the parameter values of such a dynamical system, they assume its existence).

Authors' Response:

Here, the DNB method is considered as “model free” because the three conditions of DNB were derived from the generic property of nonlinear dynamical systems. It can be proven that data generated from a nonlinear system (defined by ordinary differential equations) should satisfy the three conditions of DNB theory near the tipping point when the system state approaches the tipping point starting from a stable equilibrium. Thus, it is not necessary to infer the unknown parameters. However, as indicated by the reviewer, there may be other complicated issues due to the complexity of biological systems, and thus we removed the “model free” in the revised manuscript.

Response to the Comments of Reviewer #2

Reviewer's Comment 1:

Proofing a tumour suppressor requires endogenous genetic tests on “loss of function” and “gain of function”. Available data could only support a putative tumour suppressor at best. Similarly for metastatic suppression.

Authors' Response:

We appreciate the reviewer's valuable comment. According to your suggestion, we established a doxycycline-inducible CALML3 overexpression system in the HCC cell line HCCLM3 for gain-of-function assays and a CRISPR/Cas9 mediated knock out in the HCC cell line MHCC97L for loss-of-function assays. Those experiments both *in vitro* and *in vivo* clearly validated our conclusions of this work (Figs. 3-4 and Supplementary Figs. 3-4). In addition, our study also verified that CALML3 was a suppressor gene in HCC tumourigenesis and metastasis. In light of your suggestion, we have significantly improved our work in the revised manuscript, and the detail revisions in the manuscript are as follows.

“To further explore the role of CALML3 in HCC metastasis, we performed gain-of-function and loss-of-function studies to detect the effect of CALML3 on oncogenic behaviours of hepatoma cells including cell growth, migration and invasion. Considering that MHCC97L and HCCLM3 (metastatic potential: MHCC97L < HCCLM3) were established from the same parent HCC cell line and showed relative high and low CALML3 expression respectively, we established a doxycycline-inducible CALML3 overexpression system in the HCC cell line HCCLM3 and a CRISPR/Cas9 mediated knock out in the HCC cell line MHCC97L. Treatment with 50 ng/ml doxycycline induced a time-dependent expression of CALML3 protein in HCCLM3/CALML3 cells, detectable as early as 12 h (Fig. 3c). Two CALML3-targeting gRNAs sequences were designed to avoid non-specific effects of the CRISPR/Cas9 system. Confirmation of the genotype of two CALML3^{-/-} cell lines (CALML3^{-/-} #1 and CALML3^{-/-} #2) was testified by genomic DNA sequencing and Western blot (Supplementary Fig. 3 and Fig. 3d).

Cancer is characterized by sustaining proliferative signalling, evading growth suppressors, resisting cell death, enabling replicative immortality, inducing angiogenesis, and activating invasion and metastasis³⁸. First we examined cell proliferation ability of CALML3 overexpression or knockout cells and the

corresponding controls. The result of cell proliferation assay showed that inducible CALML3 expression by Tet-on system significantly inhibited cell growth in HCCLM3 cells, while knockout of endogenous CALML3 by CRISPR/Cas9 technology could promote cell proliferation in MHCC97L cells (Fig. 4a). The results of cell migration and invasion assay showed that compared to the controls, CALML3 overexpression cells displayed markedly decreased ability of migration and invasion, however, CALML3 knockout had an opposite effect on metastasis of HCC cells (Fig. 4b-e). In accordance with assay *in vitro*, the result of xenografts models established by inducible CALML3 overexpression system in HCCLM3 cells verified that overexpression of CALML3 significantly diminished tumorigenic capacity and pulmonary metastasis. Inducible CALML3 expression group exhibited less weight and volume of tumours, as well as less pulmonary metastasis nodules and incidence than the control (Fig. 4f-i). In addition, to further verify the role of CALML3 as a tumour suppressor, we overexpressed CALML3 in HCCLM3 cells with high metastatic potential and decreased CALML3 expression in HepG2 cells with low metastatic potential by lentiviral-mediated expression and RNA interference, respectively (Supplementary Fig. 4a). The results also showed that overexpression of CALML3 remarkably inhibited cell proliferation (Supplementary Fig. 4b), cell migration (Supplementary Fig. 4c,d), and cell invasion (Supplementary Fig. 4e,f), whereas downregulation of CALML3 had the opposite effect. We also created xenografts models of the control HCCLM3 and CALML3 overexpression HCCLM3 cells with fluorescent proteins as above (Fig. 1a); CALML3 overexpression significantly inhibited tumour growth and tumour pulmonary metastasis (Supplementary Fig. 4g,h). Together, the results of these functional assays *in vitro* and *in vivo*, suggested that CALML3 was a suppressor gene in HCC tumorigenesis and metastasis.”

”

Figure 3

Figure 3. CALML3 ranked as one of core DNB members and generation of CALML3 overexpression and knockout cells by Tet-on and CRISPR/Cas9 systems. (a) The diagram shows the DNB ranking, based on the various criteria with the four priorities, during the progression of metastasis. CALML3 was chosen as the top one for further functional tests. Note that the patterns of the change for a DNB gene at gene expression level were labelled as 1 or 0 (on the third and fourth bars). For DEG of W2-W4, C indicates a Cluster (e.g. C5 means that the gene belongs to Cluster 5). (b) The protein

expression levels of CALML3 in normal human liver epithelial cell (THLE-3) and multiple cell lines with different metastatic potentials. The rectangles with red frames showed MHCC97L, MHCC97H and HCCLM3 (metastatic potential: MHCC97L < MHCC97H < HCCLM3), which were established from the same parent HCC cell line, MHCC97. (c) Establishment and characterization of a doxycycline-inducible CALML3 overexpression system. Upper panel: Schematic of Lenti-X Tet-One inducible expression system to express inducible CALML3. Dox, doxycycline; TRE, tetracycline responsive element. Lower panel: Western analysis of CALML3 in extracts derived from inducible CALML3 cells, either untreated or induced with doxycycline (50 ng/ml) at indicated time points. (d) Generation of CALML3-knockout MHCC97L cell lines (clones of CALML3^{-/-} #1 and CALML3^{-/-} #2 cell lines) using the CRISPR/Cas9 system. di: Schematic representation of the CALML3-targeting gRNA sequences. Arrows indicate primer positions. PAM, protospacer adjacent motif. dii: Two CALML3^{-/-} cell lines were established from MHCC97L cells. The deleted sequences in the CALML3^{-/-} #1 and CALML3^{-/-} #2 cell lines are presented. Individual colonies were selected and genotyped by genomic DNA sequencing. diii: Western analysis of CALML3 in extracts derived from control and two CALML3^{-/-} cell lines. β -actin was used as the loading control.

Figure S3

Supplementary Figure 3. Individual colonies (CALML3^{-/-} #1 and CALML3^{-/-} #2 cells) of CALML3 knockout by CRISPR/Cas9 were genotyped by genomic DNA sequencing. (a) CALML3^{-/-} #1 cells had deletions of 14bp in CALML3 genomic region. (b) CALML3^{-/-} #2 cells had deletions of 11bp in CALML3 genomic region. WT, wild type.

Figure 4

Figure 4. CALML3 acts as a HCC metastatic suppressor *in vitro* and *in vivo*. (a) CALML3 overexpression in HCCLM3 cells induced by doxycycline (50 ng/ml) for indicated time inhibited cell proliferation, while CALML3 knockout by CRISPR-Cas9 technology promoted cell proliferation. Two clones (CALML3^{-/-} #1 and CALML3^{-/-} #2) targeting CALML3 were designed for functional assays. (b) CALML3 overexpression suppressed cell migration (upper panel) while CALML3 knockout brought the opposite effect (lower panel). (c) Quantification of migration cells in the indicated groups in the wound healing assay. CALML3 overexpression showed obvious suppression of migration abilities in HCCLM3 cells (upper panel) while CALML3 knockout induced significant promotion of migration abilities in CALML3^{-/-} #1 and CALML3^{-/-} #2 MHCC97L cells (lower panel). (d) CALML3 overexpression suppressed

cell invasion (upper panel) while CALML3 knockout brought the opposite effect (lower panel). **(e)** Quantification of invasive cells in the indicated groups in the transwell invasion assay. **(f)** Tumours from mice implanted with HCCLM3 cells (the control and CALML3 overexpression) in tumour-bearing mouse model. **(g)** Comparison of tumour weight and size in tumour-bearing mouse model assay. CALML3 overexpression induced by doxycycline showed less tumour weight and significantly slowed tumour growth. **(h)** Lung tissues from mice implanted with HCCLM3 cells (control and CALML3 overexpression) orthotopic transplantation model were stained with hematoxylin-eosin. **(i)** Comparison of pulmonary metastasis in xenografts models. CALML3 overexpression induced by doxycycline showed less lung metastatic nodules and incidence of lung metastasis. Dox, doxycycline. * $p < 0.05$, ** $p < 0.01$.

Figure S4

Supplementary Figure 4. CALML3 could suppress HCC metastasis *in vitro* and *in vivo*. (a) The efficiency of CALML3 overexpression in HCCLM3 cells and knockdown in HepG2 cells by lentivirus mediated expression and RNA interference was tested by Western blot. Of four different clones of CALML3 knockdown, sh4 produced the best knockdown efficiency and was chosen for further functional assays named shCALML3. (b) CALML3 overexpression suppressed cell proliferation in HCCLM3 cells while CALML3 knockdown promoted cell proliferation in HepG2 cells. (c) CALML3 overexpression suppressed cell migration (left panel) while CALML3 knockout brought the opposite effect (right panel). (d) Quantification of migration cells in the indicated groups in the wound healing assay. CALML3 overexpression showed obvious suppression of migration abilities in HCCLM3 cells while CALML3

knockdown induced significant promotion of migration abilities in HepG2 cells. (e) CALML3 overexpression suppressed cell invasion while CALML3 knockdown brought the opposite effect. (f) Quantification of invasive cells in the indicated groups in the transwell invasion assay. (g) Tumours from mice implanted with HCCLM3 cells (the control and CALML3 overexpression cells). CALML3 overexpression showed less tumour weight. (h) Lung tissues from mice implanted with HCCLM3 cells (the control and CALML3 overexpression) orthotopic transplantation model were examined by fluorescence microscopy (Left panel). CALML3 overexpression HCCLM3 cells showed less lung metastatic nodules (Middle panel) and reduced incidence of lung metastasis (Right panel). * $p < 0.05$, ** $p < 0.01$.

Reviewer's Comment 2:

All analyses are correlative results that cannot give direct and causative or mechanistic conclusions. There is NO proof that CALML3 (or any other DNB) “triggers pulmonary metastases” (in fact, it cannot be the “trigger” as explained below), hence several claims in the manuscript (including the title) are not substantiated by current data.

Authors' Response:

We appreciate your constructive suggestion to help us correct our overstatement or inaccurate statement. In light of the comment, we replaced “trigger” with “indicate” in the revised manuscript. In particular, we replaced “triggers” with “indicates” in the title. As mentioned by your comment, all analyses and functional tests (even on “loss or gain of function”) just revealed strong correlation between down-regulation of CALML3 and HCC pulmonary metastasis. It is hard to determine direct and causative conclusions without deep genetic tests due to complicated mechanisms and multi-gene dysfunctions during pulmonary metastasis.

Actually, rather than drivers, the main purpose of our study is to identify the tipping point of HCC metastasis (pulmonary metastasis) by dynamic network biomarker that is able to signal the emergence of the disease transition (qualitative change). Thus, based on your suggestion, we changed the description of CALML3 from “driver” to “indicator” of pulmonary metastases in the revised manuscript because it indicates the imminent transition to pulmonary metastases based on our data. To further explore the role of CALML3 as a suppressor in HCC metastasis in this revised manuscript, we performed gain-of-function and loss-of-function studies to detect the effect of CALML3 expression on oncogenic behaviours including cell growth, migration and invasion. The new results both *in vitro* and *in vivo* (Figs. 3-4 and

Supplementary Figs. S3-S4) suggested that CALML3 was a suppressor gene in HCC tumourigenesis and metastasis.

Reviewer's Comment 3:

Down-regulation of CALML3 may be related to EMT, but unlike Snail, from its known function, CALML3 is more likely to be a consequence than a cause ("driver" or "trigger"). Indeed in signaling, a good cancer marker needs not at all to be a "driver" or "trigger"; on the contrary, a "trigger" is often difficult to detect but the consequence is usually much easier to detect and hence can server as a better "marker".

Authors' Response:

As mentioned above, we agreed with the reviewer on the concept and meaning of "trigger" and "driver". CALML3 can play a role in indicating the tipping point of HCC pulmonary metastasis based on our functional analyses and tests. Thus, in light of the comment, we replaced "trigger" or "driver" with "indicator" in the revised manuscript. Thanks to the suggestion from the reviewer, and we will further explore the direct and causative roles of CALML3 in HCC pulmonary metastasis in future.

Reviewer's Comment 4:

As a biomarker, a regressed gene may not be a GOOD marker clinically than an amplified gene from the point-of-view of easy detection. The authors should at least compare CALML3 with existing biomarkers and discuss its advantages and disadvantages. When claiming "CALML3 is an independent predictor of HCC metastasis and recurrence", the authors should show if it is truly independent from other EMT markers, e.g. VEGF markers? There are several other similar "overstatements" or "inaccurate" minor statements that need to be revised: e.g.

Authors' Response:

Based on DNB theory, DNB members can indicate the tipping point just before the imminent transition from the observed data. In this manuscript, to study the importance of DNB members to HCC metastasis, we ranked the 334 genes in the DNB group based on comprehensive criteria which made "CALML3" as the top-one gene (although there are many other ways to choose genes). The experiment results show that "CALML3" is not only important but also novel one for HCC metastasis (pulmonary metastases) as a biomarker.

We agree with the reviewer's suggestion to compare CALML3 with existing biomarkers. We have

conducted the corresponding study, which was added in the revised manuscript. Specifically, in our study, based on IHC analyses in tumour tissue slices from patients, we show that CALML3 is a predictor of HCC metastasis and recurrence, to guide the clinical diagnosis and early treatment of HCC patients. We compared CALML3 with existing HCC metastasis related EMT markers, i.e. VEGF and vimentin by evaluating immunohistochemistry. ROC analysis was performed to evaluate the performance of CALML3, VEGF and vimentin in distinguishing patients with HCC recurrence from patients without recurrence. CALML3 AUC (0.722, 95% CI: 0.657-0.787) was higher than VEGF AUC (0.641, 95% CI: 0.572-0.711) and vimentin AUC (0.671, 95% CI: 0.604-0.739). CALML3 has a sensitivity of 70.8% and a specificity of 73.6%, higher than those of VEGF (67.8% for sensitivity and 60.4% for specificity) and vimentin (65.5% for sensitivity and 68.7% for specificity) (Supplementary Fig.5 and Supplementary Table 10). The result of ROC analysis showed that compared to VEGF and vimentin, CALML3 had a better performance of distinguishing patients with HCC recurrence from patients without recurrence. But as a suppressor gene and an indicator of HCC development, CALML3 expression was significantly reduced in HCC tumor tissues and thus was relatively harder to be detected than oncogenes with high expression. That is the disadvantage of CALML3 from the traditional viewpoint of clinical application. However, with the development of novel tools and detection techniques in future, biomarkers such as repressed genes may become easier to be detected clinically. In light of the comment, we added the descriptions in the revised manuscript.

Figure S5

Supplementary Figure 5. The predictive ability of CALML3 compared with VEGF and vimentin by receiver operating characteristic (ROC) curves for relapse-free survival (RFS). (a-c) Typical immunochemistry staining of CALML3 (a), VEGF (b) and vimentin (c) in tumour tissues of HCC patients. (d-f) ROC analysis of CALML3 (d), VEGF (e) and vimentin (f) for RFS.

Supplementary Table 10. The ROC analysis of variables for recurrence

Variables	Recurrence	
	AUC	95%CI
CALML3	0.722	0.657 to 0.787
VEGF	0.641	0.572 to 0.711
vimentin	0.671	0.604 to 0.739

Abbreviations: ROC, receiver operating characteristic; AUC, area under the curve; 95% CI; 95% confidence interval; VEGF, vascular endothelial growth factor.

Reviewer's Comment 5:

in "Discussion", "...DNB.. " is "model-free"... "without requirements on parameters or even models" which contradict with "we developed a new mathematical model..." and the three criteria serving as the parameters to be estimated from the data.

"In conclusion, our results suggest a new way to identify the tipping point or critical state with its dynamic biomarkers", this contradicts with introduction where it says DNB was already published several years ago and has been applied to several other systems, including cancer studies.

Authors' Response:

Thanks for indicating those inappropriate statements. In the revised manuscript, we have removed the "model free" and other inappropriate statements in light of your comment.

This work aims to provide the early-warning signals of pulmonary metastases by dynamic network biomarkers based on the observed data. To make this explanation accurately, the sentence "In conclusion, our results suggest a new way to identify the tipping point or critical state with its dynamic biomarkers" was replaced with "In conclusion, our results suggested a new way to identify the tipping point or critical state of HCC pulmonary metastasis with its dynamic network biomarkers, and provided biological insights into the molecular pathology of this progression from the perspectives of dynamics and network."

REVIEWERS' COMMENTS:

Reviewer #1 (Remarks to the Author):

The modifications made by the authors are satisfying for me.

Reviewer #2 (Remarks to the Author):

my concerns have been addressed completely.